# LONG-TERM FAIRNESS WITH SELECTIVE LABELS

## ABSTRACT

Long-term fairness algorithms aim to satisfy fairness beyond static and short-term notions by accounting for the dynamics between decision-making policies and population behavior. Most previous approaches evaluate performance and fairness measures from observable features and a label, which is assumed to be fully observed. However, in scenarios such as hiring or lending, the labels (e.g., ability to repay the loan) are *selective labels* as they are only revealed based on positive decisions (e.g., when a loan is granted). In this paper, we study long-term fairness in the selective labels setting, and analytically show that naive solutions do not guarantee fairness. To address this gap, we then introduce a novel framework that leverages both the observed data and a label predictor model to estimate the true fairness measure value by decomposing into the observed fairness and bias from label predictions. This allows us to derive the sufficient conditions to satisfy true fairness from observable quantities by using the confidence in the predictor model. Finally, we rely on our theoretical results to propose a novel reinforcement learning algorithm for effective long-term fair decision-making with selective labels. In semisynthetic environments, the proposed algorithm reached comparable fairness and performance to an agent with oracle access to the true labels.

## 1 INTRODUCTION

The deployment of machine learning algorithms in critical decision-making scenarios, such as admission processes (Baker & Hawn, 2022; Fuster et al., 2022) and health diagnosis, has motivated the study of algorithmic fairness. One of the most common approaches has been to demand equal benefit from a decision (e.g., acceptance in a process or correct prediction) among different demographic groups of the population (defined by race or gender, for example) (Mehrabi et al., 2021; Angwin et al., 2016). However, Liu et al. (2018) and D'Amour et al. (2020) showed that ensuring fairness at each decision round does not guarantee **fairness in the long-term** due to the feedback loop between policy deployment and the population's reaction. Furthermore, previous decisions determine the available data for policy update, which might mask decisions' unfairness.

In greater detail, long-term fairness considers that individuals are described by features $(x_t, z)$ that relate to a classification label $y_t$, and both $(x_t, y_t)$ are temporal features dependent on previous actions $a_{i<t}$ and $(x_{i<t}, y_{i<t})$. However, $z$ is a sensitive attribute, such as race or gender (considered binary in this work), and while it can be used to select actions $a_t$, the expected utility $\mu$ of the decision process should be independent of it. The disparity value $|\Delta_t| = |\mu_t^1 - \mu_t^0|$, where $\mu_t^i$ is the expected utility for the group $i$ at time $t$, serves as a measure of the unfairness of the decision process, and algorithms should satisfy that $|\Delta_t| \approx 0$ for every $t$ or when $t \to \infty$. Previous works have considered dealing with this problem with reinforcement learning algorithms (Alamdari et al., 2024; Yu et al., 2022; Yin et al., 2023; Lear & Zhang, 2025; Hu et al., 2023) and optimization approaches when the dynamics model is known (Rateike et al., 2024; Wen et al., 2021). However, these works did not consider a common characteristic of decision-making: the label $y_t$ is partially observed.

Consider the example of a loan application. The decision-maker has a binary decision to perform (approve or deny), and $y_t$ (payment ability) will only be observed in the case of acceptance. This property, called **selective labels**, presents a great impact on sequential decision-making (Bechavod et al., 2019; Ensign et al., 2018). When considering fairness, the partial observation of labels makes it non-trivial to obtain an unbiased estimate of disparity measures. Our first result (Prop. 3.1) shows that evaluating disparity only on the observed population has no guarantees over the total population.

Motivated by this negative result, in this work, we introduce a general framework for long-term fairness with selective labels, where a decision-maker leverages a label predictor $\phi$ to perform data imputation. Under this framework, we consider the difference between the true disparity $\Delta_t$ of the population and the disparity that the decision-maker sees after data imputation $\tilde{\Delta}_t$. We present a decomposition of the disparity $\tilde{\Delta}_t$ (Theo. 3.1) that relates to $\Delta_t$ by the interplay of the rejection rate of groups and the quality of predictions on the rejected population for each group. To present conditions that can be evaluated from the observed data, we introduce generalization bounds of inverse probability weighting to tackle the unknown quality of predictions on the rejected population based on the data of previously accepted individuals. Our main theoretical result (Theo 3.4) presents conditions on the observed disparity $\tilde{\Delta}_t$ and on the bias introduced by the predictor (obtained from generalization bounds) to guarantee low values of true disparity $|\Delta_t|$. Our last contribution is a novel algorithm that learns a policy and predictor model that satisfies fairness in the long-term with access to only observable quantities by satisfying the identified sufficient conditions. Based on the generalization bound, our algorithm incentivizes exploration that improves the stability of the predictor learning. Our proposed algorithm reached long-term fairness comparable to an agent with oracle access to the true disparity measure in semisynthetic environments with high-dimensional features $x_t$ and different fairness notions.

## 1.1 RELATED WORKS

For a more comprehensive discussion on related works, see Appendix A.

**Long-Term Fairness** Research in long-term algorithmic fairness has primarily leveraged reinforcement learning (RL) and causal modeling. RL solutions included model-based methods(Wen et al., 2021; Rateike et al., 2024; Deng et al., 2024), and adaptations of algorithms such as Q-learning (Alamdari et al., 2024; Chi et al., 2022), RTD3 (Yin et al., 2023) and PPO (Hu et al., 2023; Lear & Zhang, 2025; Yu et al., 2022). Prior work has defined long-term fairness either as minimizing the cumulative disparity over time (Lear & Zhang, 2025; Yu et al., 2022; Yin et al., 2023), or a long-term discounted value function of disparity (Rateike et al., 2024; Hu et al., 2023; Zhang et al., 2020a; Jabbari et al., 2017; Satija et al., 2023; Xu et al., 2024). Puranik et al. (2022) and Raab et al. (2024) introduced population dynamics through time-dependent groups occurrence. Hu & Zhang (2022) leveraged causal modeling to express the temporal dynamics between policy and population. While standard algorithms for constrained decision processes (Achiam et al., 2017; Zhang et al., 2020b) have been evaluated in long-term fairness, Rezaei-Shoshtari et al. (2024) leveraged bisimulation to transform the MDP to a new one where fairness can be satisfied without constraints. Xie & Zhang (2024); Somerstep et al. (2024) considered satisfying fairness over one iteration of strategic classification from individuals. However, all discussed solutions included the assumption that labels are available during learning.

**Selective Labels** The partial observation of data has been largely studied as selection bias. Its prevalence in standard fairness benchmarks has been highlighted by Fawkes et al. (2024). In decision-making with selective labels, prior work (Kilbertus et al., 2020; Rateike et al., 2022; Keswani et al., 2024; Chang & Wiens, 2024; Frauen et al., 2024; Lakkaraju et al., 2017) has considered an unknown but time-invariant data distribution that is sampled by the agent's policy at each iteration. To avoid exacerbating bias in this setting, (Kilbertus et al., 2020) showed that policies must explore through stochastic actions. Similarly to our approach, (Chang & Wiens, 2024) leveraged a label predictor to assess disparity in the time-invariant distribution setting. More related to this work, (Creager et al., 2020) used a causal estimator to tackle selective labels in dynamic environments. Yet, their analysis was restricted to changes occurring over a single iteration.

## 2 PRELIMINARIES AND PROBLEM FORMULATION

We consider a decision-making problem where individuals are described by features $x \in \mathcal{X}$ and a binary sensitive attribute $z \in \mathcal{Z} = \{0, 1\}$. Each individual has a latent label $y \in \{0, 1\}$, which is related to their features by the conditional probability $\alpha(x, z) := P(Y = 1 \mid X = x, Z = z)$. For each individual, the decision-maker takes a binary action $a \in \{0, 1\}$ sampled by $\pi(x, z) := P(A = 1 \mid X = x, Z = z)$ where $a = 1$ represents acceptance. In an illustrative scenario of a loan

application, $X$ might represent the financial history, $Z$ their race, $Y$ the ability to repay the loan, and $A$ the loan approval decision. For simplicity, we will assume that $X$ is a discrete feature vector; however, our results are also valid for the continuous case.

The decision-maker will select actions to maximize a reward function $R(y, a) = a(y - c)$ where $c \in \mathbb{R}^+$ represents the cost of acceptance (e.g., loan amount). Simultaneously, individuals obtain utility from the process by a function, for example, $U(y, a) = 1\{y = a\}$ where $1\{\cdot\}$ is an indicator function. The possible different definitions of $R$ and $U$ reflect the different interests the decision-maker and the applicants might have.

**Static Fairness** The decision-maker has the objective of maximizing $\mathbb{E}[R(Y, A)]$. However, in high-stakes domains, the employed policy should satisfy that utility is independent of the protected attribute (Barocas et al., 2023). A common approach to evaluate the *static fairness* of a policy is based on the disparity in the expected utility between groups: $\Delta := \mu^1 - \mu^0$, where $\mu^i := \mathbb{E}[U(Y, A)|\mathcal{C}^i]$ is the expected utility of a group $i$ with conditioning event $\mathcal{C}^i$. A fair policy $\pi$ must satisfy $|\Delta| \leq \omega$, for some small tolerance $\omega \in \mathbb{R}^+$. Different fairness notions can be expressed by the choice of $U$ and $\mathcal{C}$. In this work, we consider three common formulations: 1) *Qualification Parity* (Zhang et al., 2020a) where $\mu^i = \mathbb{E}[Y|Z = i]$; 2) *Accuracy Parity* (Berk et al., 2021) where the utility is the "accuracy" of actions $\mu^i = \mathbb{E}[1\{Y = A\}|Z = i]$; and 3) *Equality of Opportunity* (Hardt et al., 2016) where utility is the true positive rate $\mu^i = \mathbb{E}[A|Y = 1, Z = i]$.

Decision-making induces reactions from the population, where actions have an effect in future states (Perdomo et al., 2020). This motivates the consideration of features $(x_t, y_t)$ as time-dependent, commonly employing the Markov Decision Process formulation (Gohar et al., 2024).

**Definition 1** (Markov Decision Process (MDP) adapted from Wen et al. (2021))**.** A Markov Decision Process (MDP) is a tuple $\mathcal{M} := \langle S, \mathcal{A}, P_0, P_\mathcal{T}, R \rangle$ where $S$ is a set of states, $\mathcal{A}$ is a set of actions, $P_0 : S \to [0, 1]$ is the initial distribution of states, $P_\mathcal{T}(s, a, s') : S \times \mathcal{A} \times S \to [0, 1]$ is the probability of reaching state $s'$ given action $a$ at state $s$, $R : S \times \mathcal{A} \to \mathbb{R}$ is a reward function.

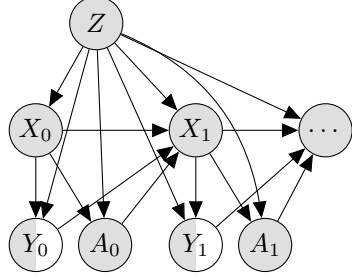

Figure 1: Graphical model for $\mathcal{F}$-MDP. $Y_t$ is partially observed depending on $A_t = 1$.

We extended the MDP definition to represent the dynamic process of decision-making by defining the state as the observable features of each individual and the transition and reward functions as dependent on the binary label.

**Definition 2** ($\mathcal{F}$-MDP)**.** An $\mathcal{F}$-MDP is a tuple $\langle S, \mathcal{A}, P_0, P_\mathcal{T}, R, U, \alpha \rangle$ where $\langle S, \mathcal{A}, P_0, P_\mathcal{T}, R \rangle$ follows the MDP definition with $S = \mathcal{X} \times \mathcal{Z}$ and $\mathcal{A} = \{0, 1\}$. Furthermore, each state has an associated label $y_t$ that follows the distribution $Y_t|X_t, Z \sim Be(\alpha(X_t, Z))$. $R$ and $U$ are the reward and utility functions, respectively, evaluated with tuples $(y_t, a_t)$.

This definition reorganizes the variables of the introductory decision-making problem into a dynamic environment (Fig. 1). In this model, the individuals' sensitive attribute $Z$ and $\alpha$ are assumed to be time-invariant, similarly to previous works (Rateike et al., 2024; Hu & Zhang, 2022). Furthermore, we consider that the action $A$ is independent of $Y$ when $X, Z$ are known. While there is no direct effect of action $a_t$ on the label $y_t$, it will influence future labels through the path $A_t \to X_{t+1} \to Y_{t+1}$.

**Long-term Fairness** In a dynamic $\mathcal{F}$-MDP, fairness constraints must be satisfied at each step. We are interested in the per-step disparity $\Delta_t = \mu_t^1 - \mu_t^0$, where $\mu_t^i := \mathbb{E}[U(Y_t, A_t) \mid \mathcal{C}^i]$. The expectation is taken over the distribution induced by policy $\pi$ and environment dynamics $(P_0, P_\mathcal{T}, \alpha)$, conditioned on $\mathcal{C}^i := \{Z = i\}$ for qualification/accuracy parity or $\mathcal{C}^i := \{A = 1, Z = i\}$ for equality of opportunity. The optimization problem is then:

$$\max_\pi \quad \mathbb{E}_{\pi, \alpha, P_\mathcal{T}, P_0} \left[ \sum_{t=1}^T R(Y_t, A_t) \right] \qquad \text{s.t.} \qquad |\Delta_t| \leq \omega \ \forall t \qquad (1)$$

This framing considers that the decision-making can steer the population towards future states where achieving fairness might have a lower cost to the reward objective.

**Selective Labels** In many practical settings, while the decision-maker observes $(X_t, Z)$ and perform decisions based on it, the true label $Y_t$ is hidden and observed only for accepted individuals $(A_t = 1)$[1]. However, as previously discussed, the information of $Y_t$ is necessary for evaluating the group-wise utilities. The reward value of an action $A_t = 0$ is always 0, independently of $Y$, however, for example, with $U(y, 0) = 1\{y = 0\}$, the utility could be either 1 or 0 if $A_t = 0$ depending on the unknown label $y$. This creates a challenge: how to evaluate fairness metrics that depend on the label $Y_t$?

## 3 MEASURING DISPARITY UNDER SELECTIVE LABELS

In this section, we study how we can employ a label predictor and quantities calculated from observed data to constrain true disparity under the selective labels scenario. Initially, we discuss the pitfalls of a simpler solution to calculate disparity.

**Disparity in the Accepted Population** Under the selective labels scenario, the decision-maker must evaluate fairness by only using labels $Y_t$ from previously accepted individuals. A naive approach is to compute the disparity only with the accepted subset of the population as follows:

$$\Delta_t^{A=1} = \mathbb{E}[U(Y_t, A_t) \mid \mathcal{C}^1, \{A_t = 1\}] - \mathbb{E}[U(Y_t, A_t) \mid \mathcal{C}^0, \{A_t = 1\}] \tag{2}$$

However, this measure is unaware of the disparity present within the rejected population.

**Proposition 3.1** (Formal presentation in Appendix B.1). *For the three fairness notions (Sec. 2), $\Delta_t^{A=1} = 0$ is not a sufficient condition to have $\Delta_t = 0$. In particular, for equality of opportunity, $\Delta_t^{A=1}$ is always 0.*

Prop. 3.1 shows that $\Delta_t^{A=1}$ is a flawed objective for learning when disparity measures are dependent on $Y_t$. A policy can be optimized to minimize $\Delta_t^{A=1}$ and learn to mask the true disparity measure. For instance, with qualification parity, the policy can learn to accept individuals who have a similar distribution of labels between groups without ensuring equal qualification in the total population.

**Imputation Model** To be able to calculate disparity from the complete population, the decision-maker can employ a model $\phi : \mathcal{X} \times \mathcal{Z} \to [0, 1]$ to predict unseen labels and evaluate fairness based on the imputed labels. We set predicted labels sampled by $\hat{Y}_t | X_t, Z \sim Bernoulli(\phi(X_t, Z))$, and define the imputed label as $\tilde{Y}_t = A_t Y_t + (1 - A_t)\hat{Y}_t$. That is, with acceptance ($A_t = 1$) the true label is used ($Y_t$) and with rejection ($A_t = 0$), we use the prediction ($\hat{Y}_t$). We then compute the observed disparity $\tilde{\Delta}_t = \tilde{\mu}_t^1 - \tilde{\mu}_t^0$, where $\tilde{\mu}_t^i := \mathbb{E}[U(\tilde{Y}_t, A_t)|\tilde{\mathcal{C}}^i]^2$ is the utility calculated using $\tilde{Y}_t$.

Due to the complexity of real-world data, predictions will not correctly classify all samples and can amplify biases due to the data availability. Following, we analyze the relation between errors from the predictor model and the distortion of true fairness.

### 3.1 DECOMPOSITION OF DISPARITY WITH A LABEL PREDICTOR

As discussed by previous works, the policy influences disparity by two paths: the direct influence from decisions at each iteration and the indirect influence from previous decisions that determined the current state (Lear & Zhang, 2025; Hu & Zhang, 2022). With our imputation model, the policy $\pi$ has an extra effect on the observed disparity $\tilde{\Delta}_t$: it sets when the predictor $\phi$ is used for data imputation. We formalize this effect in the following theorem.

**Theorem 3.1** (Observed Disparity Decomposition). *Let $\epsilon_t^i := \mathbb{E}[\hat{Y}_t - Y_t|A_t = 0, Z = i]$ and $r_t^i := P(A_t = 0|Z = i)$ be, respectively, the predictor error on the rejected population and the*

---

[1]Our framework is capable of handling two formulations of selective labels: 1) the label $Y$ is hidden but exists for every individual and 2) the label $Y$ is only realized with the acceptance and is undefined for rejected individuals.

[2]With equality of opportunity, the condition $\mathcal{C}^i = \{Z = i, Y = 1\}$ is replaced by $\tilde{\mathcal{C}}^i = \{Z = i, \tilde{Y} = 1\}$.

*rejection rate for group $i$ at time $t$. Then, the observed disparity $\tilde{\Delta}_t$ can be decomposed for each fairness notion:*

- *Qualification parity ($\tilde{\Delta}_t = \mathbb{E}[\tilde{Y}_t|Z=1] - \mathbb{E}[\tilde{Y}_t|Z=0]$): $\tilde{\Delta}_t = \Delta_t + (r_t^1\epsilon_t^1 - r_t^0\epsilon_t^0)$*

- *Accuracy parity ($\tilde{\Delta}_t = \mathbb{E}[1\{\tilde{Y}_t = A_t\}|Z=1] - \mathbb{E}[1\{\tilde{Y}_t = A_t\}|Z=0]$): $\tilde{\Delta}_t = \Delta_t - (r_t^1\epsilon_t^1 - r_t^0\epsilon_t^0)$*

- *Equality of opportunity ($\tilde{\Delta}_t = \mathbb{E}[A_t|Z=1, \tilde{Y}_t=1] - \mathbb{E}[A_t|Z=0, \tilde{Y}_t=1]$): $\tilde{\Delta}_t = \mu_t^1\kappa_t^1 - \mu_t^0\kappa_t^0$ where $\mu_t^i = \mathbb{E}[A_t|Z=i, Y_t=1]$, $\kappa_t^i = 1 - r_t^i\epsilon_t^i/\tilde{\phi}_t^i$, and $\tilde{\phi}_t^i = P(\tilde{Y}_t=1|Z=i)$.*

Theo. 3.1 shows that the observed disparity $\tilde{\Delta}_t$ is confounded by the bias on $r_t^i\epsilon_t^i$ (or $r_t^i\epsilon_t^i/\tilde{\phi}_t^i$) that relates the policy rejection rate $r_t^i$ with the predictor error $\epsilon_t^i$. When optimizing for fairness using observed data, an algorithm might inadvertently exploit the imputation bias by reducing $|\tilde{\Delta}_t|$ without improvements in $|\Delta_t|$. To avoid this, the decision-maker could obtain a bounded value of true disparity $|\Delta_t|$ by balancing the rejection rate and group error, as we show in our next result.

**Theorem 3.2** (Sufficient Conditions for Bounding True Disparity)**.** *For each fairness notion and a constant $\omega \in \mathbb{R}^+$, the following conditions are sufficient to bound the true disparity $|\Delta_t| \leq \omega$:*

- *Qualification parity and accuracy parity: $|(r_t^1\epsilon_t^1 - r_t^0\epsilon_t^0)| \leq \omega/2$ and $|\tilde{\Delta}_t| \leq \omega/2$.*

- *Equality of opportunity: $|(r_t^1\epsilon_t^1/\tilde{\phi}_t^1 - r_t^0\epsilon_t^0/\tilde{\phi}_t^0)| \leq (1-v_t)\omega/2$ and $|\tilde{\Delta}_t| \leq (1-v_t)\omega/2$ where $v_t := \max_i r_t^i\epsilon_t^i/\tilde{\phi}_t^i$.*

This theorem shows that to be able to constrain the true disparity with an upper bound of $\omega$, the uncertainty induced by the predictor error demands that the observed disparity satisfy an even lower upper bound $\omega/2$. Similarly, the imputation bias should also be constrained by $\omega/2$. The conditions for equality of opportunity are stricter when $v_t$ gets closer to 1.

However, conditions from Theo. 3.2 are not actionable, as they depend on the error in the rejected population, which has unobserved labels. To make these conditions practical, in the following section, we leverage the theory of domain adaptation to bound the error on the rejected population.

## 3.2 BOUNDING TRUE DISPARITY FROM OBSERVABLE QUANTITIES

In an iterative learning process, the decision-maker will employ a sequence of policies $\pi[1], \ldots, \pi[K]$ for $K$ iterations to perform actions and select clients. Thus, this (labeled) collected data from previous iterations can be used to estimate the error of the (unlabeled) rejected population using the theory of domain adaptation. For simplicity, we will omit the subscript $t$ in this section.

Let $A[k] \sim \pi[k]$ be the decision at iteration $k$. The feature distribution for individuals in group $i$ rejected by the current policy $\pi[K]$ is $D_R^i(x) := P(X = x|A[K] = 0, Z = i)$, and for those accepted in any iteration up to $K$ is $D_A^i(x) := P(X = x|\bigvee_{k=1}^K A[k] = 1, Z = i)$. By defining the error function $\epsilon(x, i) = \mathbb{E}[\hat{Y} - Y|X = x, Z = i]$, the error over the rejected population is $\epsilon^i = \mathbb{E}_{X \sim D_R^i}[\epsilon(X, i)]$. We can estimate this error using the accepted data via Inverse Propensity Weighting (IPW), in which from a random set of $N^i$ samples collected with $D_A^i$ we can estimate $\epsilon^i$ by $\hat{\epsilon}_{A,w}^i = \sum_{j=1}^{N^i} \epsilon(x_j, i) w(x_j, i)$ where $a[1:K]^i = P\left(\bigvee_{k=1}^K A[k] = 1|Z = i\right)$ is the acceptance rate up to iteration K and:

$$w(x, i) = \frac{D_R^i(x)}{D_A^i(x)} = \frac{a[1:K]^i}{r^i} \cdot \frac{1 - \pi[K](x, i)}{1 - \prod_{k=1}^K (1 - \pi[k](x, i))} \tag{3}$$

The weight $w(x, i)$ quantifies how much more likely the features $x$ for group $i$ are to be found in the rejected population relative to the accepted population. IPW is known for suffering from high variance whenever the denominator approaches 0 (Rateike et al., 2022); however, our approach mitigates this issue by defining $D_A^i$ as the combination of all $K$ policies obtained during training. Furthermore, we leverage generalization bounds on the IPW estimator based on the variance from Cortes et al. (2010) to present an upper bound of the error.

**Assumption 1** (Overlap). *For each group $i$, $D_R^i$ is absolutely continuous with respect to $D_A^i$.*

**Theorem 3.3** (Adapted from (Cortes et al., 2010)). *Let $d < \infty$ be the pseudo-dimension of the hypothesis space of predictor models $\phi$ and $N^i$ be the number of accepted samples for group $i$, the error on the rejected population $\epsilon^i$ for group $i$ is bounded by $\overline{\epsilon}^i$ with high probability:*

$$\epsilon^i \leq \hat{\epsilon}_{A,\mathrm{w}}^i + \mathcal{O}\left(\sqrt{d_2(D_R^i||D_A^i)}/\sqrt{N^i}\right) := \overline{\epsilon}^i \tag{4}$$

*where $d_2(D_R^i||D_A^i) = \mathbb{E}_{D_A^i}\left[\mathrm{w}(x,i)^2\right]$ is the Renyi divergence with factor 2.*

This bound permits us to be explicit about the quality of the IPW estimator of the error, which depends on the distance between distributions of rejected and accepted individuals and the number of samples $N^i$. The bound will get tighter when more data is collected and when the policy is less strict in the separation between rejected and accepted individuals. By substituting this error bound $\overline{\epsilon}^i$ in our framework, we arrive at our main practical result: *a set of fully observable and enforceable conditions for guaranteeing long-term fairness.*

**Theorem 3.4.** *For each fairness notion and a given constant $\omega \in \mathbb{R}^+$, the following conditions are sufficient to have $|\Delta| \leq \omega$ with high probability:*

- *Qualification parity and accuracy parity: $\sum_i r^i |\overline{\epsilon}^i| \leq \omega/2$, and $|\tilde{\Delta}| \leq \omega/2$.*

- *Equality of opportunity: $\sum_i r^i |\overline{\epsilon}^i|/\tilde{\phi}^i \leq (1-v)\omega/2$ and $|\tilde{\Delta}| \leq (1-v)\omega/2$, where $v = \max(r^i |\overline{\epsilon}^i|/\tilde{\phi}^i)$.*

This final theorem presents practical conditions to satisfy true fairness. It shows that an algorithm that reaches observed fairness in $\tilde{\Delta}$ can ensure true fairness $\Delta$ by two paths: 1) reduce the error bound $\overline{\epsilon}^i$ (by reducing $\hat{\epsilon}_{A,\mathrm{w}}^i$ or reducing the separation between accepted and rejected distributions) or 2) reduce the rejection rate $r^i$ of the policy $\pi$ for groups with high error bound, therefore reducing the reliance on imperfect predictions for that group.

## 4 METHOD

We present an algorithm for **SE**lective **L**abels in **L**ong-term **F**airness (SELLF) that optimizes the policy $\pi$ with regularization based on estimates of a predictor $\phi$ and promotes actions that ensure higher confidence in its estimates. We introduce a new loss term in the PPO algorithm (Schulman et al., 2017) and utilize the advantage regularization approach in (Yu et al., 2022) to constrain the policy. Simultaneously, the predictor model is learned with the data collected by PPO using IPW.

PPO is a policy gradient method for reinforcement learning capable of handling continuous state spaces. Defining the value of a state $V(s) = \mathbb{E}[\sum_t^T R(Y_t, A_t)|S_0 = s]$ and the q-value of a state, action pair $Q(s,a) = \mathbb{E}[\sum_{t=1}^T R(Y_t, A_t)|S_0 = s, A_0 = a]$, with both quantities reflecting the long-term returns, the advantage function is $A(s_t, a_t) = Q(s_t, a_t) - V(s_t)$. One of the main contributions of PPO is the clipping of the advantage to prevent gradient steps from moving the policy further away from the one from which data was collected. It uses the objective:

$$L^{PPO} = \mathbb{E}[\min(r_t(\theta_\pi)A(s_t, a_t), \mathrm{clip}(r_t(\theta_\pi), 1-\epsilon, 1+\epsilon)A(s_t, a_t))] \tag{5}$$

where $r_t(\theta_\pi) = \pi(s_t)/\pi_{\mathrm{old}}(s_t)$ sets the importance of each sample and $\epsilon$ is a clipping parameter. Furthermore, a neural network is used to approximate the value function $V$. We use the approach of advantage regularization introduced by Yu et al. (2022) to satisfy $|\tilde{\Delta}| \leq \omega/2$. The advantage function is penalized as $\hat{A}_\beta(s_t, a_t) = \hat{A}(s_t, a_t) - \beta_1 \max\{|\tilde{\Delta}_t| - \omega/2, 0\}$ with $\beta_1$ as a penalization weight. In particular, with qualification parity, we alter the penalization procedure to be based on $|\tilde{\Delta}_{t+1}|$ (replacing $\tilde{\Delta}_t$ with $\tilde{\Delta}_{t+1}$) as an action does not influence the disparity of the current iteration. The advantage will be reduced whenever $|\tilde{\Delta}_t| \geq \omega/2$.

According to Theo. 3.4, constraining $|\Delta_t|$ requires us to reduce the term $r_t^i |\overline{\epsilon}_t^i|$ (or $r_t^i |\overline{\epsilon}_t^i|/\tilde{\phi}_t^i$). To minimize it, we leverage the upper bound of $\overline{\epsilon}_t^i$ (Theo. 3.3)[3]. While the predictor error will be

---

[3]An analysis of each term in the bound from Theo. 3.3 during learning is presented in Appendix F.1.

optimized with $\phi$, the Renyi divergence term is a function of $\pi$. If the divergence becomes large, the bound on $\bar{\epsilon}_t^i$ becomes loose, the propensity weights increase, and our primary goal of reducing $|\Delta_t|$ is no longer guaranteed. This motivates using the Renyi divergence as a regularization term $L^{Renyi}$ in the combined learning objective $J(\theta_\pi) = L^{PPO} + \beta_2 L^{Renyi}$ where $c_t^i = r_t^i$ ($c_t^i = r_t^i/\tilde{\phi}_t^i$ for equality of opportunity) and:

$$L^{Renyi} = c_t^1 \hat{\mathbb{E}}[\mathrm{w}(x_t, 1)^2 | Z = 1] + c_t^0 \hat{\mathbb{E}}[\mathrm{w}(x_t, 0)^2 | Z = 0] \tag{6}$$

Moreover, we leverage data collected by PPO to train the predictor $\phi$ with binary cross-entropy loss. We employ inverse propensity weighting to adjust the distribution of samples that were collected under a selection bias imposed by $\pi$. That is, the predictor $\phi$ is optimized to minimize:

$$L^{Classif} = \sum_{i \in \{0,1\}} \mathbb{E}_{D_A^i}[\mathrm{w}(x_t, i)\ell(y_t, \phi(x_t, i))/\mathrm{w}(i)] \tag{7}$$

where $\ell$ is the binary cross-entropy evaluated at each sample and the weights $\mathrm{w}(x_t, i)$ (Eq. 3) shift the distribution to the overall distribution of individuals. To tackle the variance of IPW, we include the normalization term $\mathrm{w}(i) = \sum_{z=i} \mathrm{w}(x_t, z)$ that is used in self-normalized IPW (Swaminathan & Joachims, 2015). The pseudocode for SELLF is presented in Appendix C.

## 5 Experiments

We evaluated SELLF in semisynthetic environments, performing an ablation study of our solution and a comparison to baselines. We define the initial distribution of individuals based on real-world data and assume simple dynamics in three different environments: a loan application based on FICO scores (Liu et al., 2018), a crime recidivism based on COMPAS (ProPublica), and a new environment that simulates school admission based on ENEM (INEP, 2025) (Brazilian high school exam).

**Simulation**  To simulate the $\mathcal{F}$-MDP, we define the distributions $P_Z$, $P_0$, $P_{Y_t|X_t,Z}$, $P_{\mathcal{T}}$, and create a pool of individuals that follow the joint distribution. At each iteration, given a sampled individual $(z, x_t, y_t)$ from the pool, the decision $a_t$ is sampled from $\pi(x_t, z)$. With $(y_t, a_t)$, we calculate the reward and update the feature $x_{t+1}$ according to the modeled transition $P_{\mathcal{T}}$ and return this individual to the pool. This procedure induces the update of $P_{X_t|Z}$ to $P_{X_{t+1}|Z}$. For a detailed description of how probabilities were defined based on real datasets for each setting, we refer to Appendix D. Each agent starts with a resource of 1,000, which is updated based on obtained rewards.

**Baselines**  We compare the proposed algorithm SELLF with PPO, designed to maximize reward, two long-term fairness RL algorithms, POCAR (Yu et al., 2022) and ELBERT (Xu et al., 2024), and a constrained RL algorithm, FOCOPS Zhang et al. (2020b). For the baseline algorithms that do not consider the partial observation of features $Y$, we calculate the necessary metrics using the accepted population, similar to $\Delta^{A=1}$ (Eq. 2). We also implemented a variation of POCAR, which has oracle access to the true disparity $\Delta$, and thus serves as a reference of the attainable fairness without selective labels. To showcase the capabilities of SELLF in more realistic settings, where AS. 1 does not hold, we define SELLF (Semi-stoc.) that rejects individuals if $\pi(x, i) < 0.25$ and samples $A_t \sim Bernoulli(\pi(X_t, i))$ if $\pi(x, i) \geq 0.25$.

**Experimental Settings**  Algorithms were trained for 500,000 environment steps. Hyperparameters from PPO, which are common to all tested methods, were adopted from Yu et al. (2022). The disparity constraint was set to $\omega = 0.05$, and fairness-specific hyperparameters were tuned for each algorithm. We report results from the hyperparameter configuration that achieved the highest reward while satisfying disparity constraints. If no configuration satisfied the constraints, we report the one with the lowest disparity. Appendix E presents a complete description of the experimental procedure. In this section, we present results with a linear predictor $\phi$, and similar results with complex predictors $\phi$ are presented in Appendix F.2.

### 5.1 Lending Environment

We consider a simulated lending environment where each individual is described by a credit score $x_t \in \{1, 2, \ldots, 10\}$, with higher scores having a higher probability of repayment. At each timestep, the decision-maker can either approve or reject a loan application. If rejected, the individual's score

remains unchanged. If approved, the score increases by one upon repayment ($y_t = 1$) or decreases by one upon default ($y_t = 0$). We set the cost of acceptance as $c = 0.8$, motivated by the high cost of false positives (defaults) in lending applications. Despite being simple, this environment illustrates the inability of solutions based on static fairness to obtain fairness in the long-term (Liu et al., 2018; D'Amour et al., 2020).

**Ablation Study** We analyze the effect of the Renyi regularization (Eq. 6) on SELLF by varying the weight $\beta_2$. Using the accuracy parity fairness notion, we fixed $\beta_1 = 5$ (weight of $|\tilde{\Delta}_t|$ penalization) and evaluated $\beta_2 \in \{0, 0.01, 0.05, 0.1, 0.2\}$. Fig. 2 displays the behavior during learning of the gap between true and observed disparity, the Renyi regularization, and the final true disparity achieved by the trained agent. For values of $\beta_2 < 0.1$, the disparity gap increased during the initial training phase, ending with values higher than $0.01$. Similarly, the Renyi regularization drastically increases over time for these values of $\beta_2$. In contrast, with $\beta_2 = 0.1$ and $\beta_2 = 0.2$, the disparity gap is minimized, going to 0 as training progresses. $\beta_2 = 0.1$ also presented the lowest

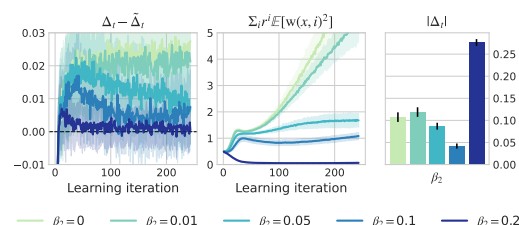

Figure 2: SELLF algorithm executed in the lending environment with $\beta_1 = 5$ and varying values of $\beta_2$. We display measures during learning and the disparity of the final policy. Results are averaged with 25 repetitions.

true disparity value among all configurations. An excessively large weight, such as $\beta_2 = 0.2$, can guide the policy to over-accept, which can present high disparity with the accuracy parity fairness notion whenever groups are not equally qualified. For that reason, $\beta_2 = 0.2$ presented the highest true disparity. This study confirmed the importance of the Renyi regularization and demonstrated that with a tuned hyperparameter, we can reach improvements in long-term fairness with selective labels.

**Comparative Results** We compare SELLF with baseline algorithms with equality of opportunity. Fig. 3 displays the behavior of trained agents for 10,000 iterations in the environment, with results summarized in Tab. 1. PPO, POCAR, ELBERT, and FOCOPS reached comparable reward, despite only PPO being designed to maximize reward. However, all four agents resulted in disparity above 0.3. As we showed in Prop. 3.1, $\Delta^{A=1}$ is a flawed objective and always 0 for equality of opportunity. For that reason, fairness-aware algorithms behave like PPO, being unable to address unfairness in the total population. SELLF and POCAR (Oracle) obtained the same disparity of 0.05 during the observed

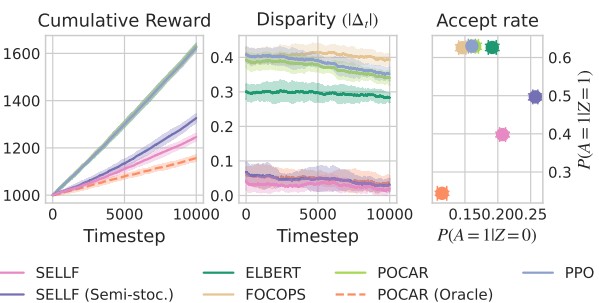

Figure 3: Reward and true disparity (equality of opportunity) over time obtained by optimized agents in the lending environment. Results are obtained with 10 repetitions. SELLF can ensure the same fairness as the baseline with oracle access and a higher reward.

period. However, SELLF was able to obtain a higher cumulative reward. This occurs as SELLF presented a higher acceptance rate than POCAR (Oracle), willing to accept individuals with higher risk to reduce the separation between accepted and rejected populations. In this setting, while SELLF (Semi-sto.) presented a higher disparity than SELLF, it was still below the constraint of 0.05. This highlights the robustness of our solution even when AS. 1 does not hold.

## 5.2 CRIME RECIDIVISM ENVIRONMENT

We present results in a crime recidivism environment based on previous works by (Zhang et al., 2020a; Rateike et al., 2024) by leveraging the COMPAS dataset (ProPublica). In this environment, individuals are described by two features, age and prior count, with $X$ having 13-dim. The sensitive attribute is $Z = 0$ if the individual is African American and $Z = 1$ if they are Caucasian. A decision-maker must choose between jail ($A = 0$) or bail ($A = 1$), and is negatively rewarded if they grant bail ($A = 1$) and the individual reoffends ($Y = 0$), and a small positive reward if $Y = 1$. We do so by setting the cost $c = 0.9$. In this example, the only dynamic present is if an individual is granted bail and reoffends, their priors count feature increases by one. In this experiment, we use the accuracy parity fairness principle. We emphasize that this environment is a simplification and does not fully represent real-world dynamics of the criminal justice system.

**Comparative Results** We display results in Fig. 4. In this environment, due to the high cost of a false positive (bail a reoffender), no algorithm obtained a final reward above $1,000$. For the same reason, POCAR (Oracle) only obtained results better than PPO when the weight of fairness penalization was increased. Our solution was the only algorithm that resulted in a disparity below $0.05$. Interestingly, SELLF (Semi-sto.) obtained the lowest disparity. However, it resulted in a lower reward than standard SELLF.

## 5.3 SCHOOL ADMISSION ENVIRONMENT

Our school admission environment is inspired by the ENEM, a Brazilian national exam. At each timestep $t$, the decision-maker selects individuals for a preparatory program and can assess the performance on the exam $y_t$ (pass/not pass) of accepted ones, while the remaining labels are unobserved. The environment dynamics are as follows: a student's probability of passing the next exam ($y_{t+1} = 1$) increases if they are selected ($a_t = 1$) or pass the current exam ($y_t = 1$). Furthermore, there is a decrease in the probability of passing the exam between timesteps due to the effect of age, which is present independently of the decision. The conditioned distribution $Y_t \mid X_t, Z$ is a logistic regression learned from data, with $X_t$ having 126 dimensions. The cost is set as $c = 0.5$. In this study, we perform experiments using the qualification parity fairness notion.

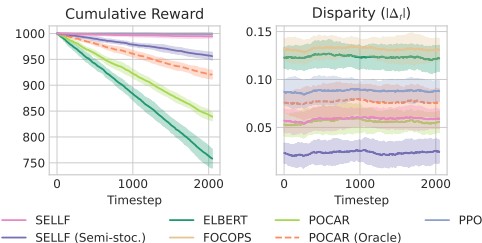

Figure 4: Reward and true disparity (accuracy) obtained in the crime recidivism environment. Results are obtained with 10 repetitions.

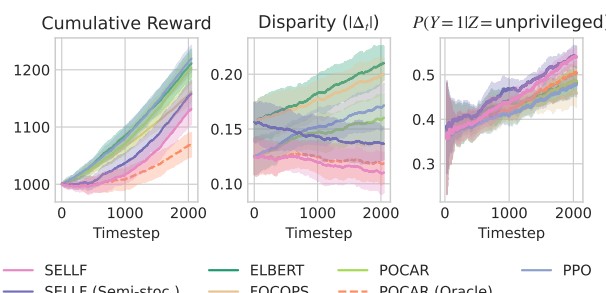

Figure 5: Reward and true disparity (qualification) obtained in the school admission environment. Results are obtained with 10 repetitions. No algorithms were able to reach a disparity below $0.05$, yet SELLF obtained the lowest values.

**Comparative Results** Fig. 5 displays the result of the trained agents. PPO, ELBERT, POCAR, and FOCOPS obtained the highest rewards. However, four algorithms ended with a disparity higher than the initial value of $0.125$. This shows that by optimizing $\Delta^{A=1}$, an agent can even cause harm in the long-term. SELLF, SELLF (Semi-sto.) and POCAR (Oracle) presented a reduction in disparity; however, not being able to reach values lower than $\omega = 0.05$. As the qualification of individuals is highly influenced by the initial state and transition dynamics, agents have less effect on it. All algorithms resulted in an increase in the qualification of the unprivileged group over time, with the highest increase obtained by SELLF.

| Model | Lending (Equal. of Opp.) | | Criminal Rec. (Acc. Parity) | | School admis. (Quali. Parity) | |
|---|---|---|---|---|---|---|
| | Disparity($\downarrow$) | Reward($\uparrow$) | Disparity($\downarrow$) | Reward($\uparrow$) | Disparity($\downarrow$) | Reward($\uparrow$) |
| PPO | 0.38 ($\pm$ 0.01) | 1624.64 ($\pm$ 14.0) | 0.09 ($\pm$ 0.01) | 998.61 ($\pm$ 2.4) | 0.15 ($\pm$ 0.01) | **1219.68 ($\pm$ 23.0)** |
| POCAR | 0.37 ($\pm$ 0.01) | 1626.68 ($\pm$ 12.9) | 0.06 ($\pm$ 0.01) | 838.94 ($\pm$ 7.9) | 0.14 ($\pm$ 0.01) | 1203.02 ($\pm$ 25.4) |
| POCAR (Oracle) | 0.05 ($\pm$ 0.01) | 1156.82 ($\pm$ 12.4) | 0.08 ($\pm$ 0.02) | 920.16 ($\pm$ 8.2) | **0.12 ($\pm$ 0.01)** | 1069.16 ($\pm$ 20.5) |
| FOCOPS | 0.41 ($\pm$ 0.01) | 1627.32 ($\pm$ 14.7) | 0.13 ($\pm$ 0.01) | 1000.0 ($\pm$ 0.0) | 0.18 ($\pm$ 0.02) | 1161.7 ($\pm$ 16.9) |
| ELBERT | 0.30 ($\pm$ 0.01) | **1630.18 ($\pm$ 13.2)** | 0.12 ($\pm$ 0.01) | 758.07 ($\pm$ 18.1) | 0.18 ($\pm$ 0.02) | 1211.36 ($\pm$ 24.2) |
| SELLF (Semi-sto.) | 0.05 ($\pm$ 0.01) | 1326.28 ($\pm$ 16.7) | **0.02 ($\pm$ 0.01)** | 956.07 ($\pm$ 7.7) | 0.14 ($\pm$ 0.02) | 1157.88 ($\pm$ 24.3) |
| SELLF (ours) | **0.03 ($\pm$ 0.01)** | 1246.24 ($\pm$ 14.7) | 0.05 ($\pm$ 0.01) | 990.72 ($\pm$ 3.5) | **0.12 ($\pm$ 0.01)** | 1131.58 ($\pm$ 26.7) |

Table 1: Performance and true disparity averaged over time of agents at the lending (with equality of opportunity) and school admission (with qualification parity) environments. Results are an average of 10 deployment repetitions.

Tab. 1 displays the average disparity and accumulated reward for agents. Additional results with varying fairness notions are presented in Appendix F. In summary, SELLF was able to obtain positive rewards while reaching fairness levels similar to an oracle in the selective labels setting.

## 6 DISCUSSION

**Assumptions**   Our theoretical analysis relies on two simplifying assumptions. First, the $\mathcal{F}$-MDP assumes stationary group dynamics. While this may not hold over extended periods, in practice, the model could be periodically retrained to adapt to new dynamics. Second, our error bounds assume overlap between the distributions of rejected and accepted individuals. That is, every individual who has a non-zero probability of being rejected also has a non-zero probability of being previously accepted. This requirement is consistent with the need for active exploration; the decision-maker must sometimes accept uncertain applicants to gather data and prevent convergence to a suboptimal policy, a principle argued by Kilbertus et al. (2020).

**Dependence on IPW**   As previously discussed, IPW can introduce high variance and learning instability if action probabilities become too small (Swaminathan & Joachims, 2015). While SELLF uses the importance weights in the Renyi and classification losses, our solution presents two safeguards to obtain reduced variance. First, the importance weights are calculated by the aggregated probability of actions from all previous policies. This cumulative probability provides a more stable denominator, preventing it from approaching zero. Second, the Renyi loss objective itself incentivizes the policy to reduce the magnitudes of weights. See Appendix F.3 for an empirical evaluation of weights.

## 7 CONCLUSION

We studied the problem of long-term fairness under selective labels. In this scenario, the decision-maker must maximize reward while satisfying fairness in regard to labels, which are only observed in the case of acceptance. We present a modeling framework based on an MDP where a predictor model is used to infer unseen labels. Under this new configuration, we presented a theoretical analysis of the relation between true and observed disparity, which was then used to motivate our proposed algorithm. By leveraging the estimates of unfairness obtained by the predictor model and a confidence bound on these estimates, we introduce a simple and flexible reinforcement-learning algorithm. In two semisynthetic environments, our algorithm presented the highest improvements in fairness, reaching similar results to an agent with oracle access to labels. Future work includes the adaptation of our theoretical results to an offline algorithm that leverages historical data, as in highly consequential settings, deploying a policy for learning might be infeasible. Furthermore, future directions also include the study of the setting in which the decision-maker selects an action among multiple possibilities (non-binary), each with different effects.

ETHICAL STATEMENT

The presented research encompasses topics of fair application of machine learning in social contexts. To evaluate our proposed algorithm, real-world data was used with sensitive information such as race and gender. Both datasets were anonymized by the source with no identifiable information available.

**LLM Usage**   The authors acknowledge the use of LLM-based tools (Gemini) as a writing assistant to improve text clarity.

REPRODUCIBILITY STATEMENT

All theoretical results presented in the main paper have their proof presented in the Appendix B. The code with the implementation of all algorithms and experiments is included as a supplemental material, with instructions on how to reproduce results. Datasets employed are open and can be downloaded from the references, and the preprocessing steps are presented in Appendix D and in the included code.

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

## A    EXTENDED RELATED WORKS

In this section, we discuss in greater detail related works on long-term fairness and selective labels, and other related areas. For a comprehensive review of long-term fairness, we refer to the survey by Gohar et al. (2024).

Long-term fairness has gained significant attention since the seminal work by Liu et al. (2018), which presented an analysis of fairness policies in a credit scenario with one-step feedback. Following D'Amour et al. (2020) employed simulations to evaluate the effect of fair policies over a larger period. Both works showed that ensuring fairness at each iteration might cause harm in the long term when dynamics are introduced.

Algorithmic solutions commonly leverage reinforcement learning solutions or causal modeling. Considering that feedback dynamics are known, Wen et al. (2021) introduced fairness metrics to the MDP setting by formulating individuals' rewards as a second objective, and Rateike et al. (2024) studied settings where a fixed-threshold policy can converge to a fair equilibrium. Works have also formulated fairness as a time-dependent cost, which is aggregated over time with discounts, defining value functions of unfairness (Satija et al., 2023; Xu et al., 2024). Particularly, Xu et al. (2024) introduced the idea of defining group-wise supply and demands, and evaluating fairness based on the ratio of such group-wise measures. Jabbari et al. (2017) analyzes a meritocratic notion of fairness in RL where an action cannot be preferred unless its long-term return is higher, highlighting the difficulty of obtaining reliable long-term value estimates. This configuration of long-term disparity also permits the use of standard algorithms for constrained MDPs (CMDPs), such as CPO (Achiam et al., 2017) and FOCOPS (Zhang et al., 2020b), which perform projections of the policy to the feasible set at each learning iteration. A set of works has studied the PPO algorithm to ensure fairness. Yu et al. (2022) and Hu et al. (2023) included a penalization term on the advantage estimate used for policy optimization, while Lear & Zhang (2025) used an expansion of the disparity in qualification as a value function. Q-learning was adapted for long-term fairness by Chi et al. (2022) and Alamdari et al. (2024).

The relation between short-term fairness and long-term fairness has also been studied by previous works (Hu et al., 2023; Alamdari et al., 2024; Lear & Zhang, 2025). Yin et al. (2023) used a different framework where states were the joint distribution of the population. To support continuous state and actions, it employed a modification of the least-squares value iteration algorithm. Deng et al. (2024) proposed learning an approximation of dynamics to evaluate if unfairness is introduced by them, that is, if a fair state can transition to an unfair state even if decisions are group-independent. A subset of works for long-term fairness considered different dynamics between decisions and population distributions, where the participation of groups was not fixed over time and depended on the quality of predictions (accuracy) or on the acceptance rates Puranik et al. (2022); Raab et al. (2024). All of these approaches considered only measuring fairness from fully observable features $X$ (no use of labels $Y$).

In the strategic classification setting, the deployment of a decision model induces the update of data distribution by the adaptation of individuals to the decision rationale. With these dynamics, Xie & Zhang (2024) studied how the iterative update of the model with strategic data could result in long-term disparity, even when fairness is enforced at each iteration. Somerstep et al. (2024) studied how such dynamics could be used to shift the data distribution to a fair state. However, works in strategic classification consider a single timestep of dynamics.

In the stochastic $K$-out-of-$N$ bandit model, the decision-maker at each iteration must select $K$ arms over $N$ total possibilities and observes rewards only for those arms. Long-term fairness has already been discussed in this setting by considering that each arm belongs to a group, and that each group should be selected (any arm of the group) with a frequency higher than a threshold Chen et al. (2020); Li & Varakantham (2022); Wang et al. (2024). While these works handle partial feedback, the classical bandit assumption that actions do not influence future contexts eliminates long-term feedback loops that motivate our work.

Partial-label scenarios have been analyzed in simpler decision-theoretic models or in settings with time-invariant data distributions. Zhang et al. (2020a) presented a theoretical study of threshold policies that satisfy fairness in the short-term, but not necessarily in the long-term. While a partial observation MDP was used in the analysis, it did not consider learning in such a setting. Fawkes

et al. (2024) audit benchmark fairness datasets and reported that selection bias (a class of bias that includes partial feedback) was identified in 85% of them. In static environments, previous works considered the problem of sequentially employing a policy that is used to learn the unseen data distribution with selective labels. Kilbertus et al. (2020) showed policies should be able to "explore" so that a learning algorithm does not end in a suboptimal utility and fairness. Following Rateike et al. (2022) considered using the unlabeled data to learn an unbiased representation of individuals' features, which were then used to train the policy. Keswani et al. (2024) presented an algorithm to learn the optimal policy with suboptimal estimates of labels. Chang & Wiens (2024) introduced a similar approach in the time-invariant setting, where a model is used to learn the treatment policy and another to learn the outcome. Their solution is an expectation-maximization algorithm that iteratively trains both models. In supervised learning, Lakkaraju et al. (2017) study selective labels in classification when data is collected from heterogeneous human decision-makers with different acceptance rates, and propose using data from lenient decision-makers to estimate failure rates of a new black-box model; however, their analysis does not consider that data is dynamically affected by decisions. Frauen et al. (2024) consider fairness in off-policy learning, aiming to learn a fair policy using data collected by a possibly discriminatory behavior policy; they assume features are static over time and learn a representation independent of the sensitive attribute while preserving predictive information, leveraging IPW similarly to our approach.

Causal modeling provides a language for defining the feedback loops that induce long-term disparity. Creager et al. (2020) discussed the benefits of representing assumptions within the causal diagrams' framework, providing various examples where an undesired effect occurs when the causal structure of the system is misspecified. One such analysis was of off-policy evaluation in the setting of partial feedback, yet their work did not include a theoretical analysis. Hu & Zhang (2022) connected causality and performative predictions in long-term fairness by transforming an optimization problem defined by a causal model to a problem of performative prediction.

## B  PROOFS

In this section, we will omit the subscript $t$ whenever it is not relevant. Furthermore, we simplify the notation $P^i(E|C) := P(E|C, Z = i)$ for any event $E$ and condition $C$. We will also write $\mathbb{E}^i[E|C] := \mathbb{E}^i[E|C, Z = i]$.

### B.1  PROOF OF PROP. 3.1

We first write the proposition presented using a formal notation.

**Proposition B.1** (Restatement of Prop. 3.1). *Let $a_t^i = P(A_t = 1|Z = i)$ be the acceptance rate of group $i$. For each fairness principle, the disparity calculated from the accepted population $\Delta_t^{A=1}$ has the decomposition $\Delta_t^{A=1} = (\mu_t^1 c^1 - \mu_t^0 c^0) + (d^1 - d^0)$ where the terms $c^i, d^i$ are:*

- *Qualification parity: $c^i = P(A_t = 1|Y_t = 1, Z = i)/a_t^i, \ \ d^i = 0$.*

- *Accuracy parity: $c^i = 1/a_t^i, \ \ d^i = -P^1(Y_t = 0, A_t = 0|Z = i)/a_t^i$.*

- *Equality of opportunity: $c^i = d^i = 0$ (that is, $\Delta_t^{A=1} = 0$ always).*

*And $|\Delta_t^{A=1}| = 0$ is not a sufficient condition for $|\Delta_t| = 0$.*

*Proof.* First, we consider each fairness principle and identify an expression for $\Delta_t^{A=1}$:

**1) Qualification parity**

By considering each term of $\Delta^{A=1}$:

$$\mu^i = P^i(Y = 1|A = 1) = P^i(Y = 1)\frac{P^i(A = 1|Y = 1)}{a^i} \tag{8}$$

And by joining both terms, we have:

$$\Delta^{A=1} = P^1(Y = 1|A = 1) - P^1(Y = 1|A = 1) \tag{9}$$

$$= P^1(Y = 1)\frac{P^1(A = 1|Y = 1)}{a^1} - P^0(Y = 1)\frac{P^0(A = 1|Y = 1)}{a^0} \tag{10}$$

**2) Accuracy parity**

Similarly, considering each side $\Delta^{A=1}$:

$$P^i(Y = A) = P^i(Y = 1, A = 1) + P^i(Y = 0, A = 0) \tag{11}$$

$$= P^i(Y = A|A = 1)a^i + P^i(Y = 0, A = 0) \implies \tag{12}$$

$$P^i(Y = A|A = 1) = \frac{P^i(Y = A)}{a^i} - \frac{P^i(Y = 0, A = 0)}{a^i} \tag{13}$$

And by joining both terms:

$$\Delta^{A=1} = P^1(A = Y|A = 1) - P^0(A = Y|A = 1) \tag{14}$$

$$= \mu^1/a^1 - \mu^0/a^0 - \left(\frac{P^1(Y = 0, A = 0)}{a^1} - \frac{P^0(Y = 0, A = 0)}{a^0}\right) \tag{15}$$

**3) Equality of opportunity**

It is straightforward to see that $P^i(A = 1|Y = 1 \wedge A = 1) = 1$, concluding that $\Delta^{A=1} = 1 - 1 = 0$ independently of the real disparity $\Delta$.

**Conclusion**

Now, if $|\Delta_t^{A=1}| = 0$ we can have $|\Delta_t| > 0$ by setting:

- Qualification parity: $c^1 \neq c^0$ and $\mu_t^1 = (c^1/c^0)\mu_t^0$ which implies $\mu^1 \neq \mu^0 \implies |\Delta_t| > 0$.

- Accuracy parity: $d^1 = d^0, c^1 \neq c^0$ and $\mu_t^1 = (c^1/c^0)\mu_t^0$ which implies $\mu^1 \neq \mu^0 \implies |\Delta_t| > 0$.

- Equality of opportunity is direct, as $\Delta_t^{A=1} = 0$ always.

$\square$

### B.2 Proof of Theo. 3.1

*Proof.* We will define the random variable $\epsilon = \hat{Y} - Y$, $\epsilon \in \{-1, 0, 1\}$ and use the relation $\tilde{Y} = Y + (1 - A)\epsilon$. Based on this, we can conclude:

$$\mathbb{E}^i[(1 - A)\epsilon] = \underbrace{\mathbb{E}^i[(1 - A)\epsilon \mid A = 1]}_{=0} a^i + \mathbb{E}^i[(1 - A)\epsilon \mid A = 0]r^i \tag{16}$$

$$= \mathbb{E}^i[\epsilon \mid A = 0]r^i = \epsilon^i r^i \tag{17}$$

With $\epsilon^i$ as defined in the section. Then, we consider each fairness principle.

**1) Equality of qualification**

Considering each term of $\tilde{\Delta}$, we have that:

$$\mathbb{E}^i[\tilde{Y}] = \mathbb{E}^i[Y + (1 - A)\epsilon] = \mathbb{E}^i[Y] + \mathbb{E}^i[(1 - A)\epsilon] \tag{18}$$

We combined both terms to rewrite $\tilde{\Delta}$:

$$\tilde{\Delta} = (\mathbb{E}^1[Y] + \epsilon^1 r^1) - (\mathbb{E}^0[Y] + \epsilon^0 r^0) \tag{19}$$
$$= (\mathbb{E}^1[Y] - \mathbb{E}^0[Y]) + (\epsilon^1 r^1 - \epsilon^0 r^0) = \Delta + (\epsilon^1 r^1 - \epsilon^0 r^0) \tag{20}$$

**2) Equality of accuracy**

Considering each term of $\tilde{\Delta}$, we have that:

$$\mathbb{E}^i[1\{A = \tilde{Y}\}] = \tag{21}$$
$$= P^i(A = 1, \tilde{Y} = 1) + P^i(A = 0, \tilde{Y} = 0) \tag{22}$$
$$= P^i(A = 1, Y = 1) + P^i(A = 0, Y + \epsilon = 0) \tag{23}$$
$$\tag{24}$$

Let's work on the term $P^i(A = 0, Y + \epsilon = 0)$:

$$P^i(A = 0, Y + \epsilon = 0) = r^i P(Y + \epsilon = 0 \mid A = 0) \tag{25}$$
$$= r^i \mathbb{E}^i[1 - (Y + \epsilon) \mid A = 0] \tag{26}$$
$$= r^i(1 - \mathbb{E}^i[Y \mid A = 0] - \epsilon^i) \tag{27}$$
$$= r^i - \mathbb{E}^i[Y \mid A = 0]r^i - \epsilon^i r^i \tag{28}$$
$$= \underbrace{r^i - P^i(Y = 1, A = 0)}_{P^i(Y=0, A=0)} - \epsilon^i r^i \tag{29}$$
$$= P^i(Y = 0, A = 0) - r^i \epsilon^i \tag{30}$$

Replacing it in $\mathbb{E}^i[1\{A = \tilde{Y}\}]$:

$$\mathbb{E}[1\{A = \tilde{Y}\} \mid Z = z^\bullet] = P^i(A = 1, Y = 1) + P^i(Y = 0, A = 0) - r^i \epsilon^i \tag{31}$$
$$= \mathbb{E}[1\{A = Y\} \mid Z = z^\bullet] - r^i \epsilon^i \tag{32}$$

Then, we have that by replacing both terms of $\tilde{\Delta}$.

$$\tilde{\Delta} = \mathbb{E}^1[1\{A = \tilde{Y}\}] - \mathbb{E}^0[1\{A = \tilde{Y}\}] = \tag{33}$$
$$= \left(\mathbb{E}^1[1\{A = Y\}] - r^1 \epsilon^1\right) - \left(\mathbb{E}^0[1\{A = Y\}] - r^0 \epsilon^0\right) = \tag{34}$$
$$= \Delta - (\epsilon^1 r^1 - \epsilon^0 r^0) \tag{35}$$

**3) Equality of opportunity**

We first open one term of $\tilde{\Delta}$:

$$\mathbb{E}^i[A = 1 \mid \tilde{Y} = 1] = P^i(A = 1 \mid \tilde{Y} = 1) = \frac{P^i(A = 1, \tilde{Y} = 1)}{P^i(\tilde{Y} = 1)} \tag{36}$$

Notice that $P^i(A = 1, \tilde{Y} = 1) = P^i(A = 1, Y = 1)$ as $\tilde{Y} = Y$ when the action is positive. We are now interested in replacing the denominator $P^i(\tilde{Y} = 1)$ to $P^i(Y = 1)$. To do so, we can define $\kappa^i = \dfrac{P^i(Y = 1)}{P^i(\tilde{Y} = 1)}$ with the assumption that $P^i(\tilde{Y} = 1) \neq 0$ and obtain:

$$\mathbb{E}^i[A = 1 \mid \tilde{Y} = 1] = \frac{P^i(A = 1, Y = 1)}{P^i(Y = 1)} \kappa^i = \mathbb{E}^i[A = 1 \mid Y = 1]\kappa^i \tag{37}$$

This shows that the true positive rate calculated from the observed labels is equal to the true positive rate with the multiplying factor $\kappa^i$, that is, the ratio of real positive labels and observed positive labels. Then, joining both terms in the expression of $\tilde{\Delta}$, we obtain:

$$\tilde{\Delta} = \mathbb{E}^1[A = 1 \mid Y = 1]\kappa^1 - \mathbb{E}^0[A = 1 \mid Y = 1]\kappa^0 \tag{38}$$

We are also interested in rewriting $\kappa^i$ to remove the direct dependence on $Y$, a value that is partially observed. We have that:

$$\mathbb{E}^i[\tilde{Y}] = \mathbb{E}^i[Y] + \mathbb{E}^i[(1 - A)\epsilon] \tag{39}$$

$$\implies P^i(Y = 1) = \mathbb{E}^i[\tilde{Y}] - \mathbb{E}^i[(1 - A)\epsilon] \tag{40}$$

$$= P^i(\tilde{Y} = 1) - \epsilon^i r^i \tag{41}$$

And then:

$$\kappa^i = \frac{P^i(\tilde{Y} = 1) - \epsilon^i r^i}{P^i(\tilde{Y} = 1)} = 1 - \frac{\epsilon^i r^i}{\tilde{\phi}^i} \tag{42}$$

With $\tilde{\phi}^i$ defined as in the section.

$\square$

## B.3 Proof of Theo. 3.2

*Proof.* We first consider the scenario of qualification parity and accuracy parity. From Theo. 3.1, we have that:

$$\tilde{\Delta} = \Delta \pm (r^1 \epsilon^1 - r^0 \epsilon^0) \implies \tag{43}$$

$$|\Delta| = |\tilde{\Delta} \pm (r^1 \epsilon^1 - r^0 \epsilon^0)| \tag{44}$$

$$\leq |\tilde{\Delta}| + |r^1 \epsilon^1 - r^0 \epsilon^0| \tag{45}$$

$$\leq \omega/2 + \omega/2 = \omega \tag{46}$$

Where the first two lines use $\pm$ due to the different expressions obtained for qualification parity and accuracy parity.

Now, with the equality of opportunity fairness principle, we have from Theo. 3.1:

$$\tilde{\Delta} = \mu^1 \kappa^1 - \mu^0 \kappa^0 \tag{47}$$

$$= \kappa^1 \Delta + \mu^0(\kappa^1 - \kappa^0) \implies \tag{48}$$

$$\kappa^1 |\Delta| = |\tilde{\Delta} - \mu^0((1 - r^1 \epsilon^1/\tilde{\phi}^1) - (1 - r^0 \epsilon^0/\tilde{\phi}^0))| \tag{49}$$

$$= |\tilde{\Delta} - \mu^0(-r^1 \epsilon^1/\tilde{\phi}^1 + r^0 \epsilon^0/\tilde{\phi}^0)| \tag{50}$$

$$\leq |\tilde{\Delta}| + \mu^0 |r^1 \epsilon^1/\tilde{\phi}^1 - r^0 \epsilon^0/\tilde{\phi}^0| \tag{51}$$

$$\leq |\tilde{\Delta}| + |r^1 \epsilon^1/\tilde{\phi}^1 - r^0 \epsilon^0/\tilde{\phi}^0| \tag{52}$$

$$\leq \frac{(1 - v)\omega}{2} + \frac{(1 - v)\omega}{2} = (1 - v)\omega \implies \tag{53}$$

$$|\Delta| \leq \frac{(1 - v)\omega}{\kappa^1} \leq \frac{(1 - v)\omega}{1 - v} = \omega \tag{54}$$

Where line 52 uses the fact that $\mu^0 \leq 1$.

$\square$

### B.4 PROOF OF THEO. 3.3

Theo. 3.3 was initially presented by Cortes et al. (2010). Here we present the original statement and describe the adaptation to our scenario. First we define models $h$, which are evaluated from a bounded loss $L(h(x), f(x))$ (abbreviated by $L_h(x)$), the risk $R(h) = \mathbb{E}_{x \sim P}[L_h(x)]$ and the weighted empirical loss $\hat{R}_w(h) = \sum_{i=1}^{m} w(x_i) L_h(x^i)$ calculated from $m$ i.i.d. samples $(x_i, y_i)$ obtained by distribution $Q$.

**Theorem B.1** (Theo. 3 from Cortes et al. (2010)). *Let $H$ be a hypothesis set such that $Pdim(\{L_h(x) : h \in H\}) = p < \infty$. Assume that $d_2(P||Q) < \infty$ and $w(x) = P(x)/Q(x) \neq 0$ for all $x$. Then, for any $\delta > 0$, with probability of at least $1 - \delta$, the following holds:*

$$R(h) \leq \hat{R}_w(h) + 2^{5/4} \sqrt{d_2(P||Q)} \sqrt[3/8]{\frac{p \log \frac{2me}{p} + \log \frac{4}{\delta}}{m}} \tag{55}$$

In our setting, we evaluated the models $\phi$ using data collected from previously accepted individuals, that is, $Q := D_A^i$ and:

$$D_A^i(x) = \frac{\left(1 - \prod_{k=1}^{K}(1 - \pi[k](x,i))\right) g(x,i)}{a[1:K]^i} \tag{56}$$

where $g(x, i) := P(X = x | Z = i)$. However, we wish to know the error from the distribution of rejected individuals, which is $P := D_R^i$:

$$D_R^i(x) = \frac{(1 - \pi[K](x,i))g(x,i)}{r^i} \tag{57}$$

and $w(x, i) = P(X)/Q(x) = D_R^i(x)/D_A^i(x)$ has the expression presented in Sec. 3. Lastly, our loss measure is $\epsilon(x, i) = \mathbb{E}[\hat{Y} - Y | X = x, Z = i]$, which is also bounded, but has support in $[-1, 1]$. With this configuration, $R(h) := \epsilon^i$ and $\hat{R}_w(h) = \hat{\epsilon}_{A,w}^i$. While the larger support changes the formulation of the bound in Eq. 55, only the constants are different, and big-O is kept the same.

### B.5 PROOF OF THEO. 3.4

*Proof.* Initially, as $\bar{\epsilon}^i \geq \epsilon^i$ and $r^i > 0, \forall i$, we have that $\sum_i r^i |\epsilon^i| \leq \sum_i r^i |\bar{\epsilon}^i|$. By leveraging results from Theo. 3.1, we have that for qualification parity and accuracy parity:

$$\tilde{\Delta} = \Delta \pm (r^1 \epsilon^1 - r^0 \epsilon^0) \implies \tag{58}$$

$$|\Delta| = |\tilde{\Delta} \pm (r^1 \epsilon^1 - r^0 \epsilon^0)| \tag{59}$$

$$\leq |\tilde{\Delta}| + |r^1 \epsilon^1 - r^0 \epsilon^0| \tag{60}$$

$$\leq |\tilde{\Delta}| + |r^1 \epsilon^1| + |r^0 \epsilon^0| \tag{61}$$

$$\leq |\tilde{\Delta}| + |r^1 \bar{\epsilon}^1| + |r^0 \bar{\epsilon}^0| \tag{62}$$

$$\leq \omega/2 + 2\omega/4 = \omega \tag{63}$$

And for equality of opportunity:

$$\tilde{\Delta} = \mu^1 \kappa^1 - \mu^0 \kappa^0 = \kappa^1 \Delta + \mu^0(\kappa^1 - \kappa^0) \implies \tag{64}$$

$$\kappa^1 |\Delta| = |\tilde{\Delta} - \mu^0(\kappa^1 - \kappa^0)| \tag{65}$$

$$\leq |\tilde{\Delta}| + \mu^0|\kappa^1 - \kappa^0| \tag{66}$$

$$\leq |\tilde{\Delta}| + |\kappa^1 - \kappa^0| = |\tilde{\Delta}| + \left| r^1 \epsilon^1 / \tilde{\phi}^1 - r^0 \epsilon^0 / \tilde{\phi}^0 \right| \tag{67}$$

$$\leq |\tilde{\Delta}| + \sum_i r^i \epsilon^i / \tilde{\phi}^i \tag{68}$$

$$\leq |\tilde{\Delta}| + \sum_i r^i \bar{\epsilon}^i / \tilde{\phi}^i \tag{69}$$

$$\leq \frac{(1-v)\omega}{2} + \frac{(1-v)\omega}{2} = (1-v)\omega \tag{70}$$

$$|\Delta| \leq \frac{(1-v)\omega}{\kappa^1} \leq \frac{(1-v)\omega}{(1-v)} = \omega \tag{71}$$

$\square$

## C  Algorithm

---
**Algorithm 1:** SELLF

---
Initialize neural networks $\pi, \phi, V$ with respective weights $\theta_\pi^0, \theta_\phi^0, \theta_V^0$ and memory buffer
$M = \{\}$;
**for** $k = 1, 2, \ldots, K$ **do**
    Initialize replay buffer $B = \{\}$;
    **for** *episode* $= 1, \ldots, E$ **do**
        **for** $t = 1, 2, \ldots, T$ **do**
            Sample $a^t \sim \pi(x^t, z), y^t \sim \alpha(x^t, z), x^{t+1} \sim P_\mathcal{T}(x^t, z, a^t, y^t), \hat{y}^t \sim \phi(x^t, z)$
            Run data imputation $\tilde{y}^t \leftarrow a^t y^t + (1 - a^t)\hat{y}^t$
            $B \leftarrow B \cup \{z, x^t, \tilde{y}^t, a^t, r^t, x^{t+1}, \tilde{\Delta}^t\}$
            $M \leftarrow M \cup \{x^t, z, \tilde{y}^t\}$ if $a^t = 1$
        **end**
    **end**
    **for** *each predictor gradient step* **do**
        Sample mini-batch from $M$
        $\theta_\phi^k \leftarrow \theta_\phi^k - \gamma \nabla_{\theta_\phi} \sum_{i \in \{0,1\}} \hat{\mathbb{E}}_{D_A^i}[\mathrm{w}(x_t, i)\ell(\tilde{y}_t, \phi(x_t, i))/\mathrm{w}(i)]$
    **end**
    **for** *each policy gradient step* **do**
        $\hat{A}_\beta(s_t, a_t) \leftarrow \hat{A}(s_t, a_t) - \beta_1 \max\{|\tilde{\Delta}_t| - \omega/2, 0\}$
        $d(\theta_\pi) \leftarrow \pi(x^t, z)/\pi_{\theta_\pi^t}(x^t, z)$
        $J^{\mathrm{CLIP}}(\theta_\pi) \leftarrow \hat{\mathbb{E}}[\min(d(\theta_\pi)\hat{A}(s_t, a_t), \mathrm{clip}(d(\theta_\pi), 1 - \epsilon, 1 + \epsilon)\hat{A}_\beta)]$
        $L^{Renyi}(\theta_\pi) \leftarrow (r_t^1 \hat{\mathbb{E}}[\mathrm{w}_t^2|Z = 1] + r_t^0 \hat{\mathbb{E}}[\mathrm{w}_t^2|Z = 0])/2$
        $\theta_\pi^{t+1} \leftarrow \theta_\pi^t + \gamma(\nabla_{\theta_\pi} J^{\mathrm{CLIP}}(\theta_\pi) - \nabla_{\theta_\pi} L^{Renyi}(\theta_\pi))$
        $\theta_V^{t+1} \leftarrow \theta_V^t - \alpha \nabla_{\theta_V} \mathbb{E}[(V(s^t) - G(s^t))^2]$ ;      $\triangleright\ G(s^t) \leftarrow \sum_{i=0}^T \gamma^i r_{t+i}$
    **end**
**end**

---

## D  Datasets and Environments

This work considers the effects of algorithms on the distribution of population attributes. This characteristic impedes the evaluation of algorithms in historical (and static) data, as they will not present the effects of the intervention of algorithms. For that reason, we employ semisynthetic

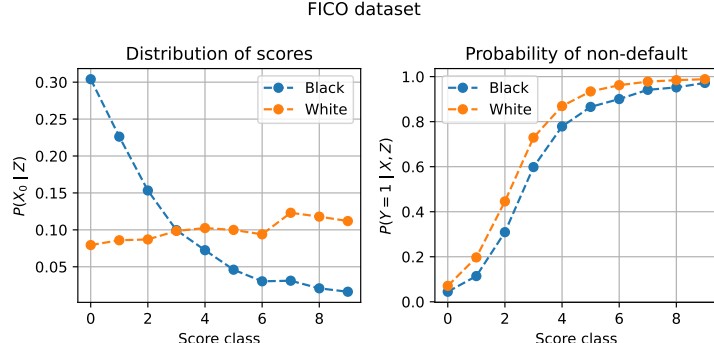

Figure 6: Probability distributions calculated from the FICO dataset to define the environment.

environments to evaluate the proposed algorithms, which is commonly done in studies of long-term fairness. The environments have initial distributions of variables $(X, Z)$ and the relation with the target $Y$ calculated from real-world datasets. To model the environment's dynamics, we assume transition functions based on features, labels, and decisions. These dynamics must be plausible for the system modeled, of which we considered three: loan applications based on FICO, crime recidivism based on COMPAS, and school admission based on ENEM.

**Lending**   FICO (Reserve, 2007) is a common open-source dataset utilized in fairness studies. It consists of anonymized profiles of clients of a banking institution with a credit score that was calculated from these attributes. Using the data available from Barocas et al. (2023), race was defined as the sensitive attribute $Z$, using two classes ("Black" and "white"). For simplicity, we set each group with probability $0.5$. Then, for each group, we calculate the probability of observing each score (from 10 possible discretized score values). This was then used as $P(X^0|Z)$. Next, we calculate the probability of payment given each score for each group, that is $\alpha(X, Z) = P(Y = 1|Z, X)$. Both distributions are presented in Fig. 6. It is possible to see that while the white population is almost uniformly spread among scores, almost 50% of the Black individuals have a score class of 0 or 1. When considering the probability of payment, we can see that both groups present very similar behavior, yet a small difference is present. We observe that the probability of payment of a Black individual of the same score class as a white one is smaller. This might be caused by external social aspects that were not fully captured by the credit score.

In the FICO environment, we used the dynamics first presented by Liu et al. (2018). If an individual is rejected, the credit score is kept the same. If an individual is accepted, their credit score will increase by one unit if $Y = 1$ and decrease by one unit if $Y = -1$.

**Criminal Recidivism**   COMPAS (ProPublica) is a software used in courts in the US to assess the likelihood of recidivism. A study of great importance by ProPublica showed that this tool consistently predicted a higher likelihood for African-Americans, indicating a discriminatory practice. In this environment, we leveraged the dataset published by ProPublica, and followed the construction of the environment as done by Zhang et al. (2020a); Rateike et al. (2024). We define two sensitive groups based on race, which are African-American and Caucasian, and use only two variables: age (discretized in 5 classes) and the number of priors (discretized in 8 classes). This results in a setting where $X$, after the one-hot encoding of variables, has a dimension of 13. The decision-maker must decide between jail ($A = 0$) and bail ($A = 1$), and is negatively rewarded if bailed individuals reoffend ($Y = 0$). We calculate the probability of recidivism from the dataset, which is depicted in Fig. 7. The distribution of $P(X|Z)$ was also obtained from the dataset. We consider a simplified dynamic, where individuals only have their features $X$ altered in the case of bail decision and recidivism, in which we move the priors count to the following group.

**School Admission**   The ENEM is a national exam applied in Brazil that serves as a score for public universities' admission process. Yearly, the data collected from applicants is shared with suitable anonymization procedures. This dataset has recently been used in fairness studies (Pereira et al., 2025; Alghamdi et al., 2022). We use this dataset to model a decision-making process where a

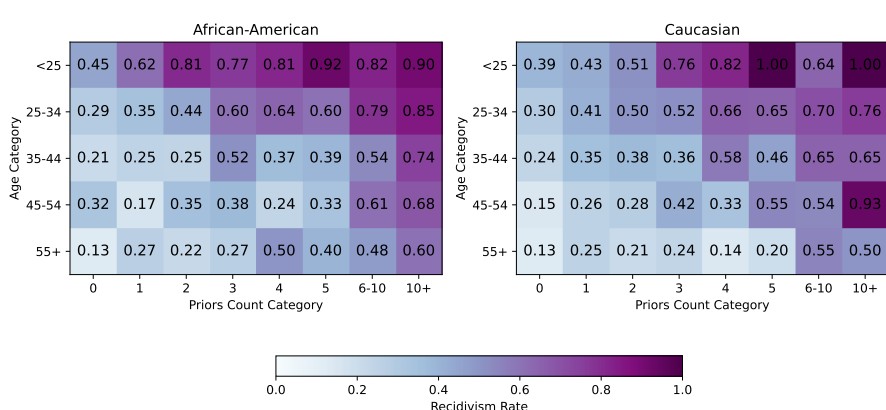

Figure 7: Probability distributions calculated from the COMPAS dataset to define the environment.

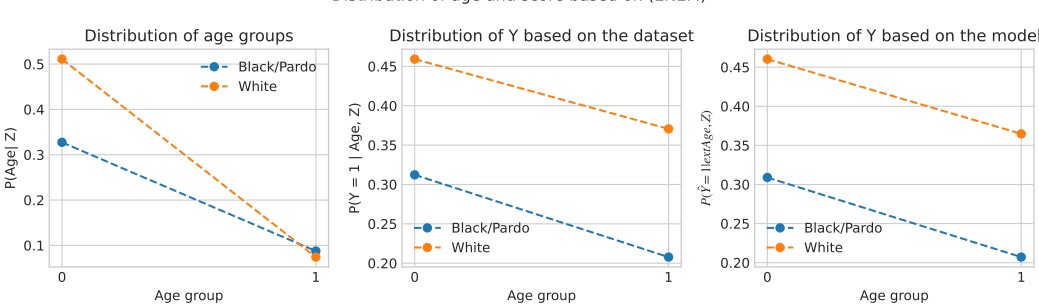

Figure 8: Estimated distributions from ENEM dataset.

decision-maker must accept/reject applicants based on attributes $X$ for a preparatory program. The label $Y$ represents reaching a grade higher than a threshold on the exam, which is only known for individuals who were participants in the program. The decision-maker has the cost of $0.4$ for an accept, and is rewarded by accepting applicants with $Y_t = 1$ (applications with $Y_t$ increase the reputation of the preparatory program). We use socioeconomic indicators as the attributes $X$. Using a random sample of 10,000 applicants from the state of São Paulo, we define $Y = 1$ if the score is higher than 575 and $0$ otherwise, which resulted in a probability of positive label of $37\%$. The sensitive attribute $Z$ is defined as the race attribute with two classes ("white" and "Black/brown") with $62\%$ and $38\%$ of occurrence, respectively. $X$ is composed of 38 categorical features which are one-hot encoded to a 126 dimensional vector.

The dynamics of this environment are defined to simulate the effect of age and of the preparatory program on $Y$. $X$ contains multiple features, one of them being a categorical age attribute with two categories (see Fig. 8 for the distribution of age categories). We consider that each iteration, the age of the applicant will increase (and other features will be kept the same), and this will affect their qualification, as displayed in the figure. We also add an extra indicator feature on the individual, that will be 1 if they have already been previously accepted or if they have previously had the label $Y = 1$, which increases the probability of the positive label by $0.5$ on the following iterations. To simulate $\alpha(X, Z)$, we fit a logistic regression from features $X, Z$ to the label $Y$. Then, whenever we updated a feature of a candidate, we used an inference with the logistic regression to obtain the probability of them having a positive label. In Fig. 8 we display the distribution of qualification among the two groups, the average qualification for each age category, and the average predicted qualification learned with the logistic model.

# E EXPERIMENTAL SETTING

**Implementation details** All algorithms and experiments were implemented using Python and Py-Torch. We followed the implementation of PPO from Stable Baselines 3[4]. The environment implementation follows D'Amour et al. (2020) and is based on Gym[5]. The algorithm POCAR was also used from the original implementation by Yu et al. (2022). The learning hyperparameters for all algorithms were as follows:

- Number of steps in data collection: 2048.
- Mini-batch size: 64.
- Epochs of policy update: 10.
- Gradient steps of predictor after data collection: 25.
- Learning rate: $10^{-5}$ for policy network and $10^{-2}$ for predictor network with exponential decay of 0.95.
- Policy $\pi$ architecture: linear layer $dim(X, Z) \times 64$, Tanh activation, linear layer $64 \times 64$, Tanh activation, linear layer $64 \times 1$. The value network has the same architecture.
- Predictor $\phi$ architecture: Linear layer $dim(X, Z) \times 1$. Sigmoid activation is used in the output to obtain probabilities.

To ensure computation efficiency, we randomly select 10 previous policies $\pi[k]$ to calculate the probability of acceptance at previous iterations.

Each algorithm was trained only once and evaluated in the environment with 10 different random seeds. Results are an average of the 10 repetitions.

**POCAR Algorithm** Yu et al. (2022) proposed advantage regularization for fairness, considering the unfairness of each (state, action) pair and the decrease of unfairness over transitions, using the following expression:

$$
\hat{A}_\beta(s_t, a_t) = \hat{A}(s_t, a_t) - \beta_1 \max\{|\Delta_t| - \omega, 0\} - \beta_2 \begin{cases} \max\{|\Delta_{t+1}| - |\Delta_t|, 0\} \text{ if } |\Delta_t| > \omega \\ 0 \text{ otherwise} \end{cases}
$$

(72)

The first term is similar to the approach used in SELLF, but it also includes a secondary term with weight $\beta_2$ that is activated whenever the disparity $|\Delta_t|$ is higher than the threshold $\omega$. This secondary term penalizes the advantage whenever the action increases the disparity from $t$ to $t+1$. This second term could also be incorporated in SELLF, but we opted to remove it for simplicity.

**ELBERT** Xu et al. (2024) formalized long-term fairness with group-wise supply $S_g$ and demand $D_g$ variables, which can be used to define common fairness metrics. For example, accuracy parity can be defined by using demand as the total number of individuals in the group and supply as the number of correct predictions between individuals for the same group. Then, the disparity measure is equal to $\Delta = S_1/D_1 - S_0/D_0$. To include the long-term effect, it also defines value functions of supply and demand, which are used to calculate a penalized advantage value on the proposed EL-BERT algorithm. As ELBERT is unaware of selection bias, we altered supply and demand variables to be calculated only within the selected population.

**FOCOPS** Proposed by Zhang et al. (2020b), FOCOPS is a constrained policy optimization algorithm inspired by TRPO. For each state, there is an associated cost (in our scenario, disparity), and a policy should satisfy constraints on the long-term discounted cost. During learning iterations, it searches for the optimal policy that satisfies constraints by projecting such a policy in the parameter space. In practice, they leverage Lagrangian weights to enforce constraints, which increase over time as constraints are violated. Similar to other baselines, FOCOPS does not consider the partial observation of labels. For that reason, we used $\tilde{\Delta}_t$ as the cost measure.

---

[4]https://stable-baselines3.readthedocs.io/en/master/

[5]https://github.com/openai/gym

---

**Algorithm 2:** Hyperparameter selection

$L_\Delta \leftarrow$ list of values $|\Delta|^{clip}$ for each hyperparameter configuration;
$L_R \leftarrow$ list of values $R_T$ for each hyperparameter configuration;
$L \leftarrow [\ ]$;
**for** $|\Delta|^{clip}, R_T$ *in* $L_\Delta, L_R$ **do**
    **if** $|\Delta|^{clip} = \min L_\Delta$ **then**
        | $L$.append($R_T$)
    **else**
        | $L$.append(0)
    **end**
    **return** *Hyperparameter configuration with highest value in L*
**end**

---

**Hyperparameters Optimization**   For POCAR and SELLF, we evaluated 12 different combinations of values of $\beta_1, \beta_2$. In both algorithms, $\beta_1$ sets the weight of the penalization of the disparity measure in the advantage, and was evaluated in $\{1, 2, 5, 10\}$. For POCAR, $\beta_2$ was evaluated in $\{1, 2, 5\}$ and for SELLF $\beta_2 \in \{0.01, 0.05, 0.1\}$. SELLF (Semi-sto.) was tuned with the same values of hyperparameters. ELBERT also includes a weight $\beta$ in the advantage penalization, in which we performed hyperparameter optimization on values $\beta \in \{1, 5, 10, 200, 2000, 20000, 200000\}$, following experiments from the original paper. FOCOPS was evaluated using the $\nu_{max}$ (the maximum value of the dual variable) in $\{2, 5, 20, 100\}$ and the long-term constraint in $\{5, 10, 15\}$. Hyperparameters of POCAR with and without oracle were tuned separately.

The selected hyperparameter configuration was the one with the highest reward that reached disparity below $\omega$ (0.05) or, if no solution reached such disparity, the one that had minimal disparity. To avoid contamination, algorithms were not given access to the true disparity measure; that is, PPO and POCAR had their hyperparameters tuned based on $\Delta^{A=1}$, POCAR (Oracle) with $\Delta$, and SELLF with $\tilde{\Delta}$. In more detail, we set $|\Delta| = \frac{1}{T} \sum_{i=1}^{T} |\Delta_t|$ (with the respective variation of the disparity measure) which was clipped $|\Delta|^{clip} = \min\{\Delta - \omega, 0\}$. Then, for each algorithm, hyperparameters were selected following Alg. 2

# F   ADDITIONAL RESULTS

In this section, we present an analysis of IPW stability during learning and results for different configurations of environments. Summarized results are presented in Tab. 3.

## F.1   ANALYSIS OF TERMS FROM THEO. 3.3

As discussed in Theo. 3.3 and detailed in Appendix B.4, the error $\epsilon_t^i$ of predictor $\phi$ on the rejected population can be bounded by $\overline{\epsilon}_t^i$, which is composed of two terms:

$$\epsilon_t^i \leq \hat{\epsilon}_{A,\text{w}}^i + 2^{5/4} \sqrt{d_2(D_R^i||D_A^i)} \sqrt[\frac{3}{8}]{\frac{p \log \frac{2N^i e}{p} + \log \frac{4}{\delta}}{N^i}} \tag{73}$$

The first term is the estimate of the error using the accepted population and the weights from IPW. The second term is an increasing function of the Renyi divergence, the pseudo-dimension $p$ of predictor space, the confidence level $\delta$, and is a decreasing function of the number of samples $N^i$. In our learning algorithm, we aim to reduce this bound by decreasing the error $\hat{\epsilon}_{A,\text{w}}^i$ and the Renyi-divergence $\sqrt{d_2(D_R^i||D_A^i)}$, as we consider that $p, \delta$ are fixed, and $N^i$ will increase over learning iterations. In this section, we present an analysis of the values of $\hat{\epsilon}_{A,\text{w}}^i, \sqrt{d_2(D_R^i||D_A^i)}$ during learning. To do so, we employ the same setting of the ablation study in Sec. 5, by using the lending environment with equality of opportunity fairness principle, $\beta_1 = 5$, and 5 different values of $\beta_2$.

We present average results over 25 random repetitions in Fig. 9, where we display the error measure and Renyi divergence of each group separately. As our results use the bias as a measure of error,

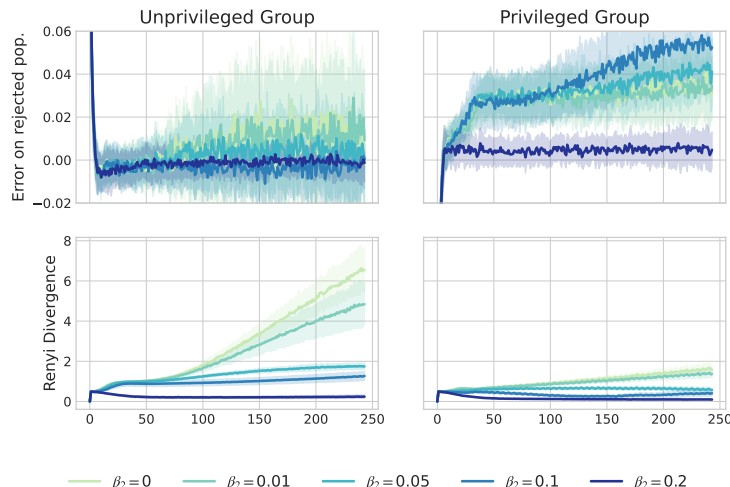

Figure 9: Error term and divergence term of the bound from Theo. 3.3. The Renyi divergence can present high values and make the bound loose.

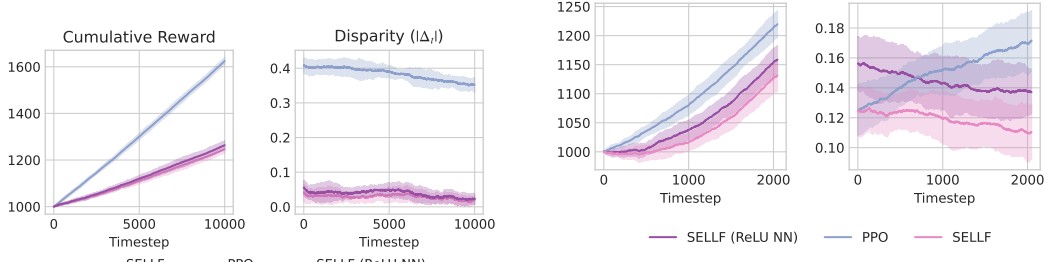

(a) Lending environment (Equality of Opportunity).

(b) School admission environment (Qualification Parity).

Figure 10: Reward and true disparity over time obtained by optimized agents. Results are obtained with 10 repetitions in the lending environment (a) and the school admission environment (b).

that is, $\epsilon(x, i) = \mathbb{E}[\hat{Y} - Y | X = x, Z = i]$ (with no module), a flexible predictor will reach values close to $0$. In the figure, we see that for both groups the error is below $0.07$ during learning, and gets close to $0$ as the value of $\beta_2$ increases. Particularly for the privileged group, we see that the bias is higher, indicating an overestimation of $Y = 1$ of the rejected population of the privileged group. When considering the Renyi divergence, our results show that it can present a significantly high value, which could make the bound loose. When the Renyi regularization is not included, that is, $\beta_2 = 0$, the Renyi divergence resulted in the highest value during learning (above $6$). These results corroborate the inclusion of the Renyi regularization in our method.

Next, we discuss and evaluate the effect of the complexity of the predictor space on SELLF.

### F.2 PREDICTOR WITH HIGH PSEUDO-DIMENSION

The bound presented in Theo.3.3 is also an increasing function of the pseudo-dimension of $\phi$, a measure of complexity of the hypothesis space. For linear predictors, as the ones evaluated in Sec. 5, this dimension is equal to $p + 1$, with $p$ being the dimension of the input size. However, for larger neural networks with ReLU activations, the pseudo-dimension will increase with the total number $W$ of parameters and number of layers $L$ by $O(WL \log W)$ (Bartlett et al., 2019). The higher the pseudo-dimension, the higher the number of samples that will be necessary to obtain a tight bound. In this section, we present results where $\phi$ is a neural network with three linear layers of hidden dimensions $[64, 64, 1]$ and ReLU activation after the first and second layers. We followed the same

| Model | Lending (Equal. of Opp.) | | School admis. (Quali. Parity) | |
|---|---|---|---|---|
| | Disparity($\downarrow$) | Reward($\uparrow$) | Disparity($\downarrow$) | Reward($\uparrow$) |
| PPO | 0.38 ($\pm$ 0.01) | **1624.64 ($\pm$ 14.0)** | 0.15 ($\pm$ 0.01) | **1219.68 ($\pm$ 23.0)** |
| SELLF | **0.03 ($\pm$ 0.01)** | 1246.24 ($\pm$ 14.7) | **0.12 ($\pm$ 0.01)** | 1131.58 ($\pm$ 26.7) |
| SELLF (ReLU NN) | 0.04 ($\pm$ 0.01) | 1263.96 ($\pm$ 17.3) | 0.14 ($\pm$ 0.02) | 1158.66 ($\pm$ 23.9) |

Table 2: Performance and true disparity averaged over time of agents at the lending (with equality of opportunity) and school admission (with qualification parity) environments. Results are an average of 10 deployment repetitions.

experimental settings from Sec. 5, performing hyper-parameter optimization on the values of $\beta_1, \beta_2$. We present a comparison of SELLF with a larger neural network, called SELLF (ReLU NN), with the implementation of SELLF with a linear predictor and the baseline PPO.

Results are displayed at Fig. 10a, 10b and Tab. 2. For the lending environment with the equality of opportunity fairness principle, both SELLF and SELLF (ReLU NN) obtained similar results, satisfying the fairness constraint of 0.05. The results obtained by SELLF (ReLU) on the school admission environment presented higher disparity than the ones obtained by SELLF. Yet, it presented a decrease in disparity over time, surpassing the PPO baseline. As $X$ has 128 features on the school admission environment, $\phi$ will contain a large number of parameters and a large pseudo-dimension, making it necessary to have a larger number of samples to reduce the bound from Theo. 3.3.

### F.3 LEARNING STABILITY

We performed a simple ablation experiment to analyze the importance weights $w(x, i) = D_R^i(x)/D_A^i(x)$ employed by SELLF, as small values of $D_A^i$ can lead to unstable learning. To do so, we evaluated the maximum value $w(x, i)$ and the minimal value of $P(A[1 : K] = 1|x, i) := P(\bigvee_{k=1}^{K} A[k] = 1|X = x, Z = i)$ during learning for different configurations of $\beta_2 \in \{0, 0.01, 0.05, 0.1, 0.2\}$ with fixed $\beta_1 = 5$. We used the lending environment with the accuracy parity fairness principle (the same one used by the ablation study in Sec. 5).

Fig. 11 presents the results of 25 random repetitions of training, with results displayed separately for each group, where 0 represents the underprivileged group. When $\beta_2 = 0$, the maximum weight of group 0 increases during learning, reaching values higher than 150 at the end. This effect is also present on the group 1, however, reaching values of 10. This difference in weights between groups occurs as they will have different acceptance rates, and the group with the lowest acceptance rate will lead to high values of importance weight. However, as we increase the value of the hyperparameter $\beta_2$, the value of $\max_x w(x, i)$ decreases for both groups, reaching very low values when $\beta_2 = 2$. This shows how the Renyi regularization can reduce the maximum value of $\max_w w(x, i)$ and consequently increase learning stability. SELLF calculates $P(A[1 : K] = 1|x, i)$ at each round by sampling 10 policies and calculating the aggregated probability of acceptance by them. With values of $\beta_2 \in \{0, 0.01\}$, this probability gets closer to 0 at the end of learning, as policies are more specialized and tend to only accept a subset of the population. When the weight of the Renyi regularization increases, this effect is reduced. While $\beta_2 = 0.1$, the probability for the unprivileged group reaches 0.2, and with $\beta_2 = 0.2$, it stays fixed at 1 after a few initial iterations.

### F.4 ENVIRONMENTS WITH OTHER FAIRNESS NOTIONS

**Lending with Accuracy Parity** Fig. 12 presents the results for accuracy parity in the lending environment. In this scenario, SELLF (Semi-sto.) obtained the best results in terms of disparity, followed by FOCOPS. In this setting, PPO was also able to obtain disparity below 0.05. This occurs as the decision-maker's utility and the individual's utility are aligned (both are positively rewarded by setting $A_t = Y_t$). All algorithms, except for POCAR (Oracle) and ELBERT, obtained similar rewards. Particularly in this environment, SELLF presented an increasing disparity over time, starting from 0.05 up to 0.08. This might occur as SELLF increases the acceptance rate of the policy to obtain better confidence bounds on $\phi$, leading to accepting individuals with $Y_t = 0$.

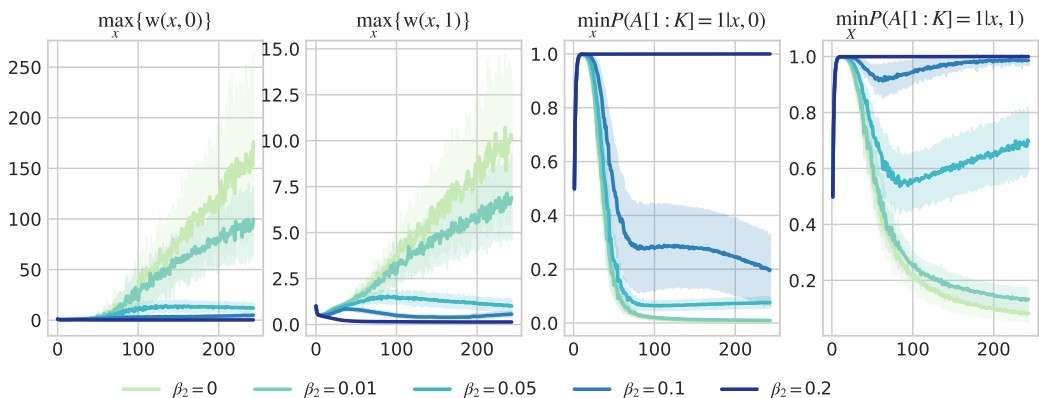

Figure 11: Behavior of importance weights $w(x, z)$ during learning and probability of acceptance at previous iterations with the lending environment with accuracy parity fairness principle.

| Model | Lending (Acc. Parity) | | Lending (Quali. Parity) | |
|---|---|---|---|---|
| | Disparity($\downarrow$) | Reward($\uparrow$) | Disparity($\downarrow$) | Reward($\uparrow$) |
| PPO | 0.04 ($\pm$ 0.01) | **1624.64 ($\pm$ 14.0)** | **0.42 ($\pm$ 0.01)** | 1607.42 ($\pm$ 15.7) |
| POCAR | 0.06 ($\pm$ 0.00) | 1611.60 ($\pm$ 15.2) | **0.42 ($\pm$ 0.01)** | 1529.86 ($\pm$ 13.3) |
| POCAR (Oracle) | 0.08 ($\pm$ 0.00) | 1417.88 ($\pm$ 21.7) | **0.42 ($\pm$ 0.01)** | 1556.70 ($\pm$ 13.5) |
| FOCOPS | 0.03 ($\pm$ 0.01) | 1617.72 ($\pm$ 16.2) | **0.42 ($\pm$ 0.01)** | 1626.5 ($\pm$ 13.8) |
| ELBERT | 0.53 ($\pm$ 0.0) | 1089.7 ($\pm$ 6.5) | **0.42 ($\pm$ 0.01)** | **1627.86 ($\pm$ 17.8)** |
| SELLF (Semi-sto.) | **0.02 ($\pm$ 0.01)** | 1627.1 ($\pm$ 16.1) | **0.42 ($\pm$ 0.01)** | 1484.32 ($\pm$ 28.8) |
| SELLF (ours) | 0.07 ($\pm$ 0.01) | 1617.54 ($\pm$ 20.5) | **0.42 ($\pm$ 0.01)** | 1611.66 ($\pm$ 29.2) |
| Model | Recidivism (Eq. Opp.) | | Recidivism (Quali. Parity) | |
| | Disparity($\downarrow$) | Reward($\uparrow$) | Disparity($\downarrow$) | Reward($\uparrow$) |
| PPO | 0.09 ($\pm$ 0.01) | 998.61 ($\pm$ 2.4) | **0.13 ($\pm$ 0.01)** | 997.9 ($\pm$ 2.3) |
| POCAR | 0.03 ($\pm$ 0.0) | 998.72 ($\pm$ 1.8) | **0.13 ($\pm$ 0.01)** | 907.46 ($\pm$ 8.9) |
| POCAR (Oracle) | 0.04 ($\pm$ 0.01) | 999.15 ($\pm$ 1.4) | **0.13 ($\pm$ 0.01)** | 998.8 ($\pm$ 1.1) |
| FOCOPS | **0.0 ($\pm$ 0.0)** | **1000.0 ($\pm$ 0.0)** | **0.13 ($\pm$ 0.01)** | **999.91 ($\pm$ 0.3)** |
| ELBERT | 0.45 ($\pm$ 0.01) | 758.07 ($\pm$ 18.1) | 0.14 ($\pm$ 0.01) | 798.17 ($\pm$ 15.1) |
| SELLF (Semi-sto.) | **0.0 ($\pm$ 0.0)** | 999.41 ($\pm$ 0.9) | **0.13 ($\pm$ 0.01)** | 919.73 ($\pm$ 10.1) |
| SELLF | 0.04 ($\pm$ 0.0) | 999.29 ($\pm$ 1.6) | **0.13 ($\pm$ 0.01)** | 997.15 ($\pm$ 2.7) |
| Model | School admis. (Eq. Opp.) | | School admis. (Acc. Parity) | |
| | Disparity($\downarrow$) | Reward($\uparrow$) | Disparity($\downarrow$) | Reward($\uparrow$) |
| PPO | 0.27 ($\pm$ 0.02) | **1211.26 ($\pm$ 13.9)** | 0.06 ($\pm$ 0.01) | 1211.26 ($\pm$ 13.9) |
| POCAR | 0.27 ($\pm$ 0.02) | 1210.78 ($\pm$ 14.3) | 0.07 ($\pm$ 0.01) | 1117.5 ($\pm$ 15.4) |
| POCAR (Oracle) | 0.05 ($\pm$ 0.02) | 1139.36 ($\pm$ 13.6) | **0.05 ($\pm$ 0.01)** | 1179.5 ($\pm$ 15.4) |
| FOCOPS | 0.05 ($\pm$ 0.01) | 1212.0 ($\pm$ 16.3) | 0.09 ($\pm$ 0.02) | 1201.58 ($\pm$ 14.6) |
| ELBERT | 0.31 ($\pm$ 0.02) | 1226.94 ($\pm$ 16.8) | 0.08 ($\pm$ 0.01) | **1226.94 ($\pm$ 16.8)** |
| SELLF (Semi-sto.) | 0.06 ($\pm$ 0.01) | 1189.4 ($\pm$ 14.1) | **0.03 ($\pm$ 0.01)** | 1205.96 ($\pm$ 18.6) |
| SELLF (ours) | **0.04 ($\pm$ 0.01)** | 1161.36 ($\pm$ 26.3) | 0.05 ($\pm$ 0.01) | 1193.28 ($\pm$ 18.6) |

Table 3: Performance of agents at the lending, recidivism, and school admission environments. Results are an average of 10 deployment repetitions.

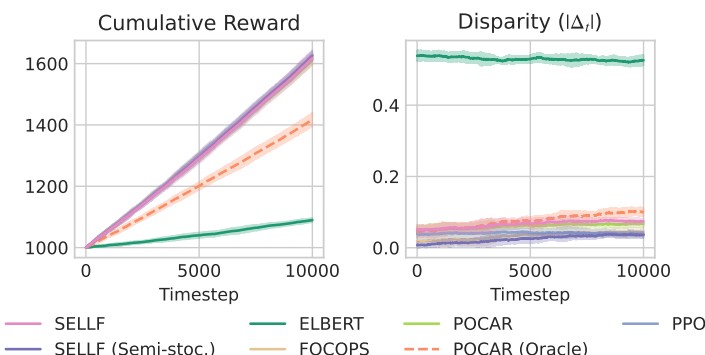

Figure 12: Reward and true disparity (accuracy parity) over time obtained by optimized agents in the lending environment. Results are obtained with 10 repetitions.

**Lending with Qualification Parity**   This environment presents a high initial unfairness of $0.43$, and considering the model $\alpha(x, z)$ as presented in Sec. D, accepting individuals with lower scores will lead to decreasing their qualification, as individuals with credit score lower or equal than $2$ have more than 50% chance of having $Y_t = 0$. For that reason, no agent was able to present improvements in terms of disparity, including POCAR with oracle access. Interestingly, SELLF obtained a reward higher than PPO in this environment. This might occur due to the incentive for acceptance introduced by the Renyi regularization.

**Criminal Recidivism with Equality of Opportunity**   In this environment, both SELLF (Semi-sto.) and FOCOPS obtained $0$ of disparity and $1000$ of reward (the same amount as the starting value). This occurs due to the high costs of false positive decisions; both algorithms resulted in an agent that has a $0$ acceptance rate. This conservative decision-making might not be ethical in this scenario, as it denies the bail opportunity for every individual. SELLF, POCAR, and POCAR (Oracle) all obtained disparity below $0.05$, with similar reward values.

**Criminal Recidivism with Qualification Parity**   Similar to the lending environment with qualification parity, all algorithms reached a similar high-disparity. This occurs as the initial disparity is considerably high, and the decision-maker does not have a significant impact on qualifications. PPO, POCAR (Oracle), FOCOPS, and SELLF obtained similar rewards in this setting.

**School Admission with Equality of Opportunity**   Fig. 16 presents the results for the school admission environment with equality of opportunity. POCAR (Oracle), SELLF and SELLF (Semi-sto.) were able to reach disparity values lower than $0.05$, with the lowest value obtained by SELLF. When considering the cumulative reward, SELLF (Semi-sto.) presented slightly higher results than POCAR (Oracle).

**School Admission with Accuracy Parity**   Fig. 17 presents the results for the school admission environment with accuracy parity. In this setting, PPO, ELBERT, FOCOPS, and SELLF (Semi-sto.) reached similar results in terms of cumulative reward, with ELBERT having the highest value. As previously discussed, the accuracy parity notion is a utility measure that behaves similarly to the reward of the decision-maker. For that reason, PPO reached a disparity measure of $0.06$. POCAR (Oracle) and SELLF presented similar results, having disparity values equal to $0.05$, but were both surpassed by SELLF (Semi-sto.). Despite this variation not having the theoretical guarantees, it consistently presented performance in terms of disparity and reward, similar to the standard variation of SELLF.

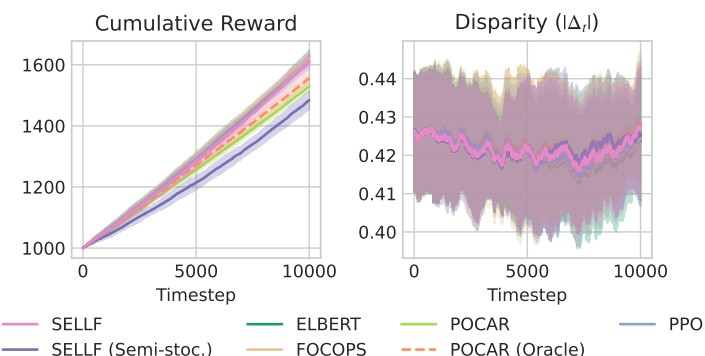

Figure 13: Reward and true disparity (qualification parity) over time obtained by optimized agents in the lending environment. Results are obtained with 10 repetitions.

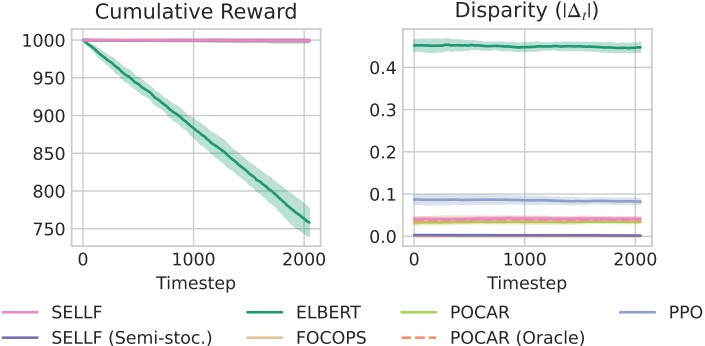

Figure 14: Reward and true disparity (equality of opportunity) over time obtained by optimized agents in the recidivism environment. Results are obtained with 10 repetitions.

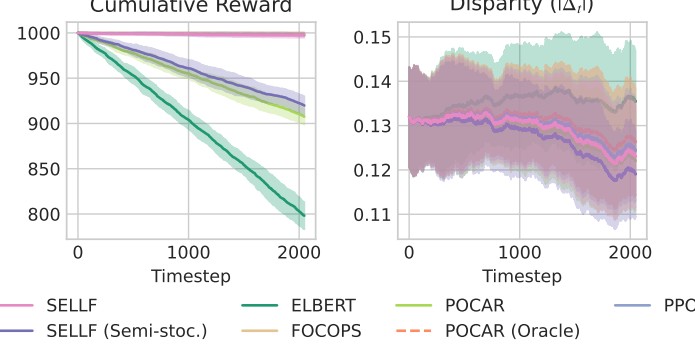

Figure 15: Reward and true disparity (qualification parity) over time obtained by optimized agents in the recidivism environment. Results are obtained with 10 repetitions.

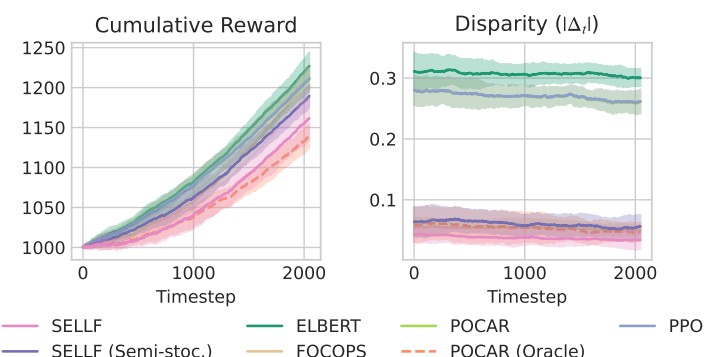

Figure 16: Reward and true disparity (equality of opportunity) over time obtained by optimized agents in the school admission environment. Results are obtained with 10 repetitions.

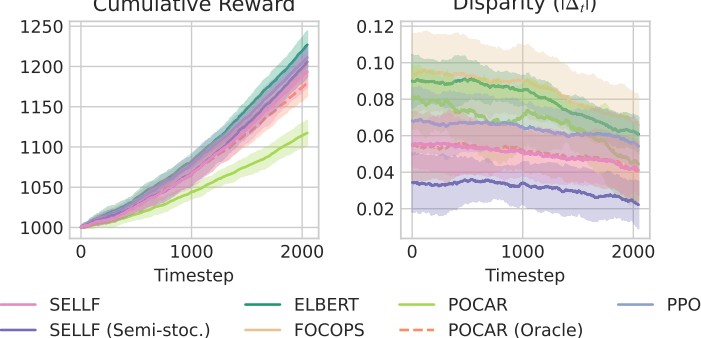

Figure 17: Reward and true disparity (accuracy parity) over time obtained by optimized agents in the school admission environment. Results are obtained with 10 repetitions.

