# OpenReview forum: "Long-term Fairness with Selective Labels"
_ICLR.cc/2026/Conference — Submitted to ICLR 2026_

### Official Review · Reviewer_xA3L · 2025-10-15

**Soundness:** 3
**Presentation:** 3
**Contribution:** 2
**Rating:** 4
**Confidence:** 3

**Summary:**

This paper studies long-term fairness in a sequential decision setting where only the labels of admitted samples are available. The paper first proves that only ensuring fairness in the observed samples cannot guarantee overall fairness. Motivated by this, the authors proposed a RL-based algorithm with an added $L^{Renyi}$ term to upper bound the fairness divergence. Experiments on synthetic datasets demonstrate the effectiveness of the algorithm.

**Strengths:**

1. The motivation is reasonable, and the selection biases do exist in real-world settings.
2. The decomposition of the disparity and the upper bound derivation seem to be correct.
3. The algorithm design is clear, and experimental results support the claims.

**Weaknesses:**

1. The method heavily depends on the accuracy of the label predictor $\phi$. But $\phi$ itself is only correct under the overlap assumption that accepted and rejected samples share enough support. In real-world settings such as loan application, it is reasonable to believe some applicants will never be accepted,i.e., some features will never be covered.

2. $\phi$ is also implemented as a simple logistic regression model, and the generalization to complex, high-dimensional data is uncertain.

3. Synthetic experiments may not reflect the applicability in real-world settings.

4. Minor: I feel that this work is closely related to fairness in sequential strategic classification and performative prediction settings, while some related works are missing (e.g., [1,2])

[1] Xie, Tian, and Xueru Zhang. "Automating data annotation under strategic human agents: Risks and potential solutions." Advances in Neural Information Processing Systems 37 (2024): 127436-127482.

[2] Somerstep, Seamus, Ya'acov Ritov, and Yuekai Sun. "Algorithmic fairness in performative policy learning: Escaping the impossibility of group fairness." Proceedings of the 2024 ACM Conference on Fairness, Accountability, and Transparency. 2024.

**Questions:**

I feel that $c$ can be very important for the performance of the algorithm. Did the authors explore how the cost affects the results?

---

> ### Author Response · Authors · 2025-11-21
> **Response to reviewer xA3L**
>
> We thank the reviewer for acknowledging the clarity of the algorithm and the strength of our empirical results. We believe that with the experiments introduced based on reviewers' suggestions, we are able to showcase the applicability of our proposal in more realistic settings. We kindly ask the reviewer to reconsider the score of our work.
>
> ## Robustness of Assumption 1
>
> To demonstrate the robustness of SELLF when the overlap assumption is violated, we have added a variation of it that does not have access to all data in Sec. 5. We introduced a hard constraint where individuals with scores lower than 0.25 are never accepted, while individuals with predicted acceptance probabilities above 0.25 are accepted by stochastic decisions. Even under this violation, SELLF ensures long-term fairness with similar values to the variation that does not include such thresholding. While our theoretical results showed the sufficiency of the overlap assumption, a direction of future work is the investigation of its necessity.
>
> ## Predictor complexity
>
> We agree that model capacity is an important consideration. To address this, we included in Appendix F.2 an evaluation of SELLF using a predictor with a larger neural network architecture. We discuss that while the higher complexity space increases the bound from Theo 3.3, the empirical performance remains strong. In the lending environment, results were the same, and in the school admission environment, the increased predictor resulted in a small decrease in long-term fairness.
>
> ## Synthetic experiments
>
> Synthetic experiments are a valid concern in fairness research. However, we emphasize that on-policy reinforcement learning requires the collection of trajectories based on the algorithms’ decisions, which is not possible using historical data. For this reason, we follow the common protocol in the literature to leverage real-world data to model the initial distribution of individuals, while assuming simplified dynamics for the feedback loop. While off-policy methods are a promising path for future work, they introduce separate challenges that are outside the scope of this paper.
>
> ## Missing references
>
> We thank the reviewer for these references. We have updated the related works (Sec. 1.1) and Appendix A to explicitly discuss the relationship between our work and fairness in sequential strategic classification.
>
> ## Acceptance cost
>
> Thank you for raising this point. We want to clarify that c is a configuration of the environment rather than a model hyperparameter. High costs incentivize higher rejection rates, exacerbating selection bias and making fairness harder to achieve. In the current version of the manuscript, we deliberately chose high cost values ($c = 0.8$ for Lending, $c=0.9$ for Recidivism) to test our algorithm in harder settings. In preliminary experiments during development, the standard implementation of PPO was able to satisfy a disparity below $0.05$ when using lower costs, such as $c=0.5$.

---

> > ### Comment · Reviewer_xA3L · 2025-11-27
> >
> > Thanks for the rebuttal. I will increase my score to 6.

---

### Official Review · Reviewer_RQL1 · 2025-10-31

**Soundness:** 2
**Presentation:** 3
**Contribution:** 3
**Rating:** 4
**Confidence:** 4

**Summary:**

This paper studies the problem of satisfying long-term fairness in scenarios with selective labels, that is, where action feedback is only observable based on positive decisions. The authors argue that this specific setting can create a flawed objective where optimal policies can learn to minimize $\Delta^{A=1}_t$ without necessarily minimizing disparity within the rejected population. They propose a framework for approaching these problems and an algorithm to solve it based on advantage regularization of PPO. Finally, they conduct experiments on 2 case studies and show that they are able to achieve higher reward and lower disparity than an oracle baseline.

**Strengths:**

* The paper studies an important problem in considering long-term fairness with partial observability on rejected candidate labels. The context the authors study this in is high-stakes and can have a significant effect on people's lives.
* The proposed method is fairly simple to implement since it is just regularizing the PPO advantage.
* The experimental results support the claims made by the authors and their proposed framework/algorithm.

**Weaknesses:**

* There seems to be some highly related works [1, 2] not mentioned in the paper. Could the authors please give a comparison with these works?
* It would be a more compelling paper to include at least one more case study (ideally studying Accuracy Parity now, since the other two proposed fairness formulations are studied by the two given case studies), and more baseline methods, to compare with SELLF. For example, [1] considered the tasks of criminal justice, health care, and insurance. Another example could be long-term exposure fairness in recommendation [4]. For baselines, [1] provided a contraction method to address this problem. Another potential baseline is an constrained RL approach such as FOCOPS [2] or CPO [3] and treat the disparity as an expected cost to minimize.

I am happy to raise my score if my concerns above are addressed.

Some minor Typos (probably run a typo checker upon revision):
* Line 104: selection -> select
* Line 211: cofounded -> confounded
* Line 228 -> depends -> depend
* Line 286: Labes -> Labels?
* etc.


[1] Lakkaraju, H., Kleinberg, J., Leskovec, J., Ludwig, J., & Mullainathan, S. (2017). The Selective Labels Problem: Evaluating Algorithmic Predictions in the Presence of Unobservables. Proceedings of the 23rd ACM SIGKDD International Conference on Knowledge Discovery and Data Mining, 275–284. Presented at the Halifax, NS, Canada. doi:10.1145/3097983.3098066

[2] Zhang, Y., Vuong, Q., & Ross, K. W. (2020). First Order Constrained Optimization in Policy Space. arXiv [Cs.LG]. Retrieved from http://arxiv.org/abs/2002.06506

[3] Achiam, J., Held, D., Tamar, A., & Abbeel, P. (2017). Constrained Policy Optimization. arXiv [Cs.LG]. Retrieved from http://arxiv.org/abs/1705.10528

[4] Mansoury, M., & Mobasher, B. (2023). Fairness of Exposure in Dynamic Recommendation. arXiv [Cs.IR]. Retrieved from http://arxiv.org/abs/2309.02322

[5] Chang, T., & Wiens, J. (07 2024). From Biased Selective Labels to Pseudo-Labels: An Expectation-Maximization Framework for Learning from Biased Decisions. Proceedings of Machine Learning Research, 235, 6286–6324.

[6] Yu, E. Y., Qin, Z., Lee, M. K., & Gao, S. (2022). Policy optimization with advantage regularization for long-term fairness in decision systems. Proceedings of the 36th International Conference on Neural Information Processing Systems. Presented at the New Orleans, LA, USA. Red Hook, NY, USA: Curran Associates Inc.

**Questions:**

Please see weaknesses. Also, some additional questions:
* I notice you use a linear predictor architecture. Does increasing the number of layers have any effect on your performance?
* Line 136: Should the beta distribution $Be(\cdot)$ require 2 parameters, $\alpha, \beta$, instead of just the one you provided?
* Is there a reason why you put $L^{\text{Renyi}}$ into the objective function in Line 109, rather than creating an additional regularization term in the advantage? I am wondering if placing this penalty term in the objective vs advantage will incur any reward hacking issues as seen in [6].
* In Line 326, what is semisynthetic about the environments?

---

> ### Author Response · Authors · 2025-11-21
> **Response to reviewer RQL1**
>
> We thank the reviewer for acknowledging the significance of our problem and the strength of our empirical results. Based on the reviewers' comments, we introduced a new use case with crime recidivism, two new baselines, and an experiment with a predictor with multiple layers. We kindly ask the reviewer to consider increasing the score.
>
> ## Missing baselines
>
> Following the reviewer’s suggestion, we implemented FOCOPS [2] and ELBERT [3] (long-term fairness baseline). The updated results show that neither baselines can ensure fairness over the total population due to ignoring the partial observation of labels. However, we do not include a comparison with [1], as it presents a different setting where there is access to historical data from multiple decision-makers.
>
> ## New case study
>
> We have introduced a third experimental environment based on crime recidivism (COMPAS) using the accuracy parity fairness principle. Consistent with the other environments, SELLF successfully reduces long-term disparity, highlighting the robustness of the method across different domains and fairness definitions.
>
> ## Missing related works
>
> Based on all reviewers' suggestions, we updated the section of related works (Sec 1.1) and Appendix A to include missing references.
>
> ## Predictor complexity
>
> The linear predictor has advantages due to the reduced pseudo-dimension, as it will require a smaller number of samples to ensure a tight bound of Theo 3.3.  We introduced an experiment in Appendix F.2 to discuss this aspect. In it, SELLF was able to surpass baselines even with a larger predictor.
>
> ## Advantage regularization
>
> During development, we considered introducing a secondary regularization term on the advantage function. In our preliminary analysis, results were highly sensitive to the weighting configuration of both terms.
>
> ## Minor clarifications
>
> - We apologize for the confusion; $Be()$ was intended to be a Bernoulli distribution. We correct this notation.
> - The initial data distribution of individuals is obtained from real-world datasets; however, dynamics are assumed. We update Sec. 5 to clarify this.
>
> [1] Lakkaraju, H., Kleinberg, J., Leskovec, J., Ludwig, J., & Mullainathan, S. (2017). The Selective Labels Problem: Evaluating Algorithmic Predictions in the Presence of Unobservables. Proceedings of the 23rd ACM SIGKDD International Conference on Knowledge Discovery and Data Mining, 275–284. Presented at the Halifax, NS, Canada. doi:10.1145/3097983.3098066
>
> [2] Zhang, Yiming, Quan Vuong, and Keith Ross. "First order constrained optimization in policy space." Advances in Neural Information Processing Systems 33 (2020): 15338-15349.
>
> [3] Xu, Yuancheng, et al. "Adapting Static Fairness to Sequential Decision-Making: Bias Mitigation Strategies towards Equal Long-term Benefit Rate." International Conference on Machine Learning. PMLR, 2024.

---

> > ### Comment · Reviewer_RQL1 · 2025-11-25
> >
> > Thank you for the detailed rebuttal and additional experiments! A few comments:
> > - Figure 10a seems to be missing `SELLF (ReLU NN)` results?
> > - Should still run through a typo checker at some point, e.g. Line 168 "hight" -> high
> >
> > Otherwise, my original concerns have been addressed and I have raised my score.

---

> > > ### Author Response · Authors · 2025-11-27
> > >
> > > We thank the reviewer for raising the score and providing further feedback. We updated Figure 10a to correctly display SELLF (ReLU NN) and carefully read the paper to correct grammar and typos.

---

### Official Review · Reviewer_rN9Y · 2025-11-03

**Soundness:** 2
**Presentation:** 2
**Contribution:** 2
**Rating:** 4
**Confidence:** 3

**Summary:**

This paper tackles the critical problem of achieving long-term fairness in sequential decision-making systems where the true outcome (label) is only observed for selections. This "selective labels" setting is common in real-world scenarios like lending (where repayment ability is only known if a loan is granted) or hiring (where job performance is only known for hired candidates). The authors first demonstrate formally that naive approaches, such as measuring fairness only on the sub-population with observed labels, are insufficient and can fail to guarantee fairness for the overall population. Then the paper introduces a novel theoretical framework that uses a label predictor to impute the labels for the "rejected" population. The core of their theoretical contribution is a decomposition (Theo. 3.1) that precisely links the true disparity to the observed disparity calculated using the predictor's imputed labels. This decomposition shows that the observed disparity is confounded by the policy's rejection rate and the predictor's error on the rejected group.

To address the problem, the authors propose a new reinforcement learning algorithm, SELLF (SElective Labes in Long-term Fairness). SELLF is based on PPO and incorporates the paper's theoretical insights through two key mechanisms: (1) It penalizes the observed disparity; and (2) It introduces a novel regularization term ($L^Renyi$) that penalizes the Renyi divergence. This new loss term directly corresponds to the theoretical bounds and encourages the policy to take actions that reduce the predictor's error and improve the confidence of its estimates.

**Strengths:**

- The paper's primary originality lies in its formal problem formulation to model the intersection of long-term fairness and selective labels.

- The authors provide a clear, formal progression from demonstrating the failure of naive methods to a full decomposition of the observed disparity in Theorem 3.1 and to actionable, observable conditions in Theorem 3.4.

- The mathematic framework of this paper is rigorous. The authors have shown how did they identify the disparity, and its error bounds directly motivates the design of the proposed learning algorithm.

**Weaknesses:**

- The motivation for choosing Inverse Propensity Weighting (IPW) to estimate the predictor's error on the rejected population ($D_R^i$) using data from the accepted population ($D_A^i$) feels abrupt. This justification is critical because the entire framework and algorithm are now built upon IPW, which is notoriously unstable and suffers from high variance, especially when acceptance probabilities are low. And I would suggest the authors to provide some implication on Assumption 1.
- The paper's core premise of *selective labels* is a potentially problematic and imprecise way to frame the problem. A more accurate conceptualization would be data selection bias. The issue is not necessarily that the label $Y$ is selectively realized only upon a positive action $A\_t=1$. Rather, a true, latent qualification $Y$ should be assumed to exist for all individuals, and a separate selection variable $S$ (in this work, the policy's action, $A_t$, which is based on the model's predictions) merely determines whether $Y$ is observed by the decision-maker. The data selection is standard in missing data and causal inference literature.
- The introduction of the $L^\text{Renyi}$ loss in Eq. (6) is confusing. The loss is justified as a practical way to control the theoretical error bound from Theorem 3.3. However, this relies on the unsubstantiated assertion that this specific Renyi divergence term "will be dominated" by other terms in the bound. This makes the $L^\text{Renyi}$ term feel like a complex and indirect proxy, obscuring the direct connection between the algorithm's objective and the actual goal of minimizing fairness disparity.
- The framework depends on a label predictor ($\phi$) to estimate the unobserved error ($\epsilon_t^i$). However, this predictor is itself trained on the same selectively-labeled, biased data (Eq. 7). The paper does not fully address how errors or biases in this predictor (resulting from unstable IPW training) might in turn corrupt the disparity estimate ($\tilde{\Delta}$) and the error bounds ($\bar{\epsilon}^i$).
- Although the paper is framed as a "long-term" fairness study, the objective (Eq. 1) is to satisfy a static fairness constraint $|\Delta_t| \le \omega$ at every timestep $t$. This formulation does not fully capture the dynamics of fairness over time, such as how unfairness at one step might be permissibly traded for greater fairness at a later step. The experiments show the results of fairness over time, but the problem's objective remains a per-step constraint rather than a truly holistic long-term objective.

**Questions:**

1. Line 50:  presents great in impact in sequential decision-making -> presents a great impact on sequential decision-making
2. Line 52: make it not trivial to obtain -> make it non-trivial to obtain
3. Line 62: *presents* conditions
4. Line 104: The decision-maker will *select* actions to maximize a reward function.
5. In Definition 2, where does $\mathcal{F}$ of $\mathcal{F}$-MDP come from? And what is the notation of $Be(\cdot)$?
6. As shown in Figure 1, the selection variable $A$ is only dependent on the features $X$ and $Z$. Should not this also depend on the true qualification $Y$?
7. Line 228: as they depends on -> as they depend on
8. Line 256: In the first sentence of Theorem 3.3, Let $d < \infty$ be the psuedo-dimension

---

> ### Author Response · Authors · 2025-11-21
> **Response to reviewer rN9Y 1/3**
>
> We thank the reviewer for acknowledging the progression of our results and the rigor of our mathematical framework. We appreciate the detailed critique regarding IPW stability and problem framing. We have addressed these concerns as follows:
>
> ## Dependence on IPW
>
> We agree that IPW estimators can suffer from high variance, particularly when overlap is poor. However, SELLF is based on two mechanisms designed specifically to avoid variance:
>
> 1. Instead of applying IPW to calculate the ratio of distributions only for the current policy, we calculate IPW as the ratio of the current policy rejecting an instance by the probability of accepting such an instance at any training iteration. That allows us to define an IPW that treats all training data as the source distribution, thereby avoiding features with low observation probability as the number of training iterations increases. We have updated Sec. 3.2 to better discuss this aspect.
>
> 2. The Renyi regularization (previously Renyi loss) explicitly penalizes the divergence between the distributions, acting as a stabilizer for the IPW estimator. By penalizing the divergence between the policy and the data distribution, we explicitly prevent the policy from drifting into regions where propensity weights would explode.
> We empirically validate this in Appendix F.3, showing that the maximum propensity weights remain small throughout training when Renyi regularization is applied, effectively neutralizing the instability typically associated with IPW in this setting.
>
>
> Regarding Assumption 1, we empirically validated the method's robustness by testing a scenario where this assumption is violated. In Sec. 5, we applied a hard rejection threshold ($<0.25$) during learning and deployment, and individuals with probability of acceptance higher than the threshold are accepted stochastically. Despite this violation, SELLF maintained superior long-term fairness compared to baselines. This confirms that while the assumption is useful for theoretical guarantees, the algorithm remains effective in more realistic settings.
>
>
> ## Selective labels
>
> Our work was developed considering the case where Y is a hidden label that is always present, but only revealed under acceptance. As we discuss in the footnote at Sec. 2, our method can be generalized to the scenario where Y is only realized under acceptance; however, it is not the main focus of our work. We update Sec. 2 to clarify our formulation.
>
> ## Renyi regularization
>
> We have updated Sec. 4 and added Appendix F.1 to clarify that the Renyi term is not an arbitrary proxy, but has the objective of minimizing the bound from Theo. 3.3. Since the error on the rejected population is unobservable (due to missing labels), we cannot minimize it directly. Theo. 3.3 proves that the observable error plus the Renyi Divergence constitutes a theoretical upper bound on this unobservable error. By minimizing the Renyi term, we are making this bound tighter. Our ablation study (Sec. 5.1) confirms this: removing this term loosens the bound, causing a significant divergence between the observed and true disparity.

---

> > ### Author Response · Authors · 2025-11-27
> > **Response to reviewer rN9Y 2/3**
> >
> > ## Training of the predictor
> >
> > This circular dependency is what SELLF addresses through the Renyi regularization. Without this regularization, the policy is not penalized for severely separating the accepted and rejected population, which could create a "vicious cycle" of bias where the predictor never sees data from regions of the feature space. To minimize the Renyi divergence, the policy tends to keep higher acceptance rates for feature values that have low probability of being observed in previous training iterations. This ensures the agent continues to accept a diverse set of individuals, collecting the unbiased labels necessary to correct the predictor. In our new experiment at Appendix F.1, we show that errors remain small for both groups, and with the introduction of the Renyi regularization, we can reduce the prediction error on the unprivileged group.
> >
> > ## Definition of long-term fairness
> >
> > In the literature, works have considered two formulations of long-term fairness: “fairness at the final state”  [2] and “fairness during the process” [1].  While our formulation was designed to focus on constraints applied at every step, we believe that the procedure of data imputation and the decomposition from Theo. 3.1  could be employed to build a value function of disparity and is an interesting direction of future work. However, as advocated by [3], practical applications of algorithm decision-making should be able to satisfy fairness at every iteration, not only over a large time horizon.
> >
> > [1] Yu, Eric Yang, et al. "Policy optimization with advantage regularization for long-term fairness in decision systems." Proceedings of the 36th International Conference on Neural Information Processing Systems. 2022.
> >
> > [2] Hu, Yaowei, and Lu Zhang. "Achieving long-term fairness in sequential decision making." Proceedings of the AAAI Conference on Artificial Intelligence. Vol. 36. No. 9. 2022.
> >
> > [3] Alamdari, Parand A., et al. "Remembering to Be Fair: Non-Markovian Fairness in Sequential Decision Making." International Conference on Machine Learning. PMLR, 2024.
> >
> > ## Minor clarifications
> >
> > - The symbol $\mathcal F$ is just a stylistic choice, and $Be()$ was intended to represent the Bernoulli distribution. We correct this notation.
> > - The action at time $t$ is not dependent on the variable $Y$, as we assume that the label is hidden and only revealed after acceptance. For that reason, the decision-maker cannot leverage this information to select the action.

---

> > > ### Author Response · Authors · 2025-11-27
> > > **Response to reviewer rN9Y 3/3**
> > >
> > > We are writing to let you know that we have just updated our response to clarify the discussion of the use of IPW and the Renyi regularization. As the discussion period is coming to an end, we remain available if you have any final questions or need further clarification.

---

### Official Review · Reviewer_5xkx · 2025-11-03

**Soundness:** 2
**Presentation:** 2
**Contribution:** 3
**Rating:** 6
**Confidence:** 4

**Summary:**

This paper studies long-term fairness by addressing the selective labels problem, where outcomes are only observed for positively decided cases. The authors show that naive approaches to measuring fairness under selective labels fail to guarantee true population-level fairness. They introduce a framework that uses a label predictor to impute missing labels and derive theoretical conditions under which observed disparity bounds translate to true disparity bounds. Experiments are conducted on lending and school admission environments.

**Strengths:**

1. The combination of long-term fairness and selective labels is both relevant and, to the best of my knowledge, previously understudied. The authors correctly point out the gap and address it to an extent.
2. The theoretical contributions are strong (disclaimer: I have not thoroughly verified the proofs).
3. The proposed method has a principled algorithmic design grounded in the theoretical results and can handle multiple fairness notions.

**Weaknesses:**

1. Assumption 1, that every feature combination with non-zero rejection probability must also have non-zero acceptance probability, is very strong and potentially unrealistic. In practice, certain subpopulations may be systematically excluded. The authors should better acknowledge these limitations. While I understand these assumptions are needed for the theoretical results, the authors could improve the paper’s practical relevance by experimentally evaluating cases where these assumptions do not hold. The stationarity assumption of the F-MDP is only briefly discussed at the end; further investigation of its implications in practice would also be valuable.

2. Some aspects of the algorithmic design are unclear. The paper moves from bounding predictor error to minimizing it using Renyi divergence as a proxy. In Line 307, the authors state that “in practice, the bound from Theo. 3.3 will be dominated by the divergence term,” but no proof or empirical validation supports this claim. Why not directly optimize for the prediction error?

3. The baseline selection is quite limited and POCAR is the only fair RL algorithm compared against. I suggest including at least two additional baselines from the list below. To my knowledge, Xu et al. (2023) is a particularly strong comparison.



4. Some citations are missing; I suggest the authors include the following (the list is not exhaustive):
* Jabbari, Shahin, et al. "Fairness in reinforcement learning." International conference on machine learning. PMLR, 2017.
* Satija, Harsh, et al. "Group fairness in reinforcement learning." Transactions on Machine Learning Research (2023).
* Xu, Yuancheng, et al. "Adapting static fairness to sequential decision-making: Bias mitigation strategies towards equal long-term benefit rate." arXiv preprint arXiv:2309.03426 (2023).
* Rezaei-Shoshtari, Sahand, et al. "Fairness in Reinforcement Learning with Bisimulation Metrics." arXiv preprint arXiv:2412.17123 (2024).
* Deng, Zhihong, et al. "What hides behind unfairness? exploring dynamics fairness in reinforcement learning." Proceedings of the Thirty-Third International Joint Conference on Artificial Intelligence. 2024.
* Frauen, Dennis, Valentyn Melnychuk, and Stefan Feuerriegel. "Fair off-policy learning from observational data." Proceedings of the 41st International Conference on Machine Learning. 2024.

**Questions:**

Please answer my questions above.

---

> ### Author Response · Authors · 2025-11-21
> **Response to reviewer 5xkx**
>
> We thank the reviewer for acknowledging the novelty of addressing selective labels in long-term fairness and for highlighting the strength of our theoretical contributions. We have addressed the concerns regarding assumptions, baselines, and algorithmic clarity as follows.
>
> ## Robustness of Assumption 1
>
> To address concerns regarding the reliance of SELLF on Assumption 1, we updated our experiments (Sec. 5 and Appendix F.4). We implemented a variant, ”SELLF (Semi-sto.)”, which enforces a hard rejection threshold of 0.25 during learning and in deployment. That is, individuals with a probability of acceptance below 0.25 will be rejected, and those with a higher probability will be accepted stochastically. Even under this violation of Assumption 1, SELLF retains high performance and fairness, outperforming all baselines. While Assumption 1 is useful for theoretical derivation, SELLF is robust and effective in realistic, threshold-based settings where other methods fail.
>
> ## Motivation of Renyi regularization
>
> To clarify the motivation behind the Renyi Regularization (previously Renyi Loss), it is essential to view it as the connection between two models: the policy and the predictor. In the selective labels setting, the predictor's error on the rejected population is unobservable, and while the IPW estimator can be leveraged, its proximity is dependent on the Renyi divergence, as shown in Theo. 3.3. We updated Sec. 4 to better discuss this motivation.
>
> To provide empirical validation, we added Appendix F.1, which tracks the magnitude of both terms in the bound (Theo 3.3) during training. We observe that the predictor error term remains consistently small ($<0.1$), whereas the Renyi divergence term increases to 7.0 when left unregularized ($\beta_2 = 0$). This empirically confirms that the Renyi divergence is indeed the dominant factor in the error bound, validating the necessity of minimizing it directly to control the error on the rejected population.
>
> ## Additional baselines
>
> We have implemented two additional baselines in Sec. 5: ELBERT (for long-term fairness) [2] and FOCOPS [1] (constrained RL). The results (summarized in Tab. 1 and Tab. 3) show that, as both algorithms rely on observed data without correcting for selective labels, they fail to minimize the true disparity, not being able to surpass the performance of SELLF. Additionally, we included an extra case study with a crime recidivism environment with the accuracy parity fairness principle, in which SELLF also obtained the best results in terms of long-term fairness.
>
> ## Missing citations
>
> We thank the reviewer for the comprehensive list. We have expanded the related works section (Sec. 1.1)  to include these references and discussed them with more detail in Appendix A.
>
> ## Conclusion
>
> We believe that the experiments with violated assumption 1 and the additional baselines directly address the concerns regarding the practical application of SELLF. If you find these results convincing, we kindly ask that you consider increasing your score. We are happy to provide any additional information.
>
>
> [1] Zhang, Yiming, Quan Vuong, and Keith Ross. "First order constrained optimization in policy space." Advances in Neural Information Processing Systems 33 (2020): 15338-15349.
>
> [2] Xu, Yuancheng, et al. "Adapting Static Fairness to Sequential Decision-Making: Bias Mitigation Strategies towards Equal Long-term Benefit Rate." International Conference on Machine Learning. PMLR, 2024.

---

> > ### Author Response · Authors · 2025-11-27
> >
> > We are writing to let you know that we have just updated our response to address your feedback more clearly. As the discussion period is coming to an end, we remain available if you have any final questions.

---

### Author Response · Authors · 2025-11-21
**Summary of Improvements**

# Summary of Improvements

We thank the reviewers for their constructive feedbacks, and for acknowledging the novelty and rigor our work;  “_relevant and, to the best of my knowledge, previously understudied_” (reviewer 5xkx), “_mathematic framework of this paper is rigorous_” (reviewer rN9Y), and that empirical results support our proposal; “_experimental results support the claims_” (reviewer xA3L) and “_the experimental results support the claims made by the authors_” (reviewer RQL1).

While we appreciate these compliments, we took the reviewers' concerns seriously. Addressing the raised issues not only improved the manuscript but also provided an opportunity to demonstrate that our method performs reliably even under stricter conditions. In summary, we have updated the paper to include:

- Inclusion of missing references in related works discussions (Sec. 1.1 and Appendix A).
- Emphasis on the safeguards of our proposal to ensure IPW stability  (lines 259 and Appendix F.3) and its relation to Renyi regularization (lines 314 and Appendix F.1).
- Inclusion of two extra baselines, ELBERT [2] and FOCOPS [1], in experiments (Sec. 5 and Appendix F.4), and a third usage scenario with crime recidivism (Sec. 5.2).
- Experiments on the robustness of our proposal in a configuration where the overlap assumption does not hold (Sec. 5) and when a complex neural network is used as the predictor (Appendix F.2).

We believe these extensive updates bridge the gap between positive comments and the quantitative scores, and we kindly ask reviewers to reconsider the assessment of our paper.



[1] Zhang, Yiming, Quan Vuong, and Keith Ross. "First order constrained optimization in policy space." Advances in Neural Information Processing Systems 33 (2020): 15338-15349.

[2] Xu, Yuancheng, et al. "Adapting Static Fairness to Sequential Decision-Making: Bias Mitigation Strategies towards Equal Long-term Benefit Rate." International Conference on Machine Learning. PMLR, 2024.

---

### Author Response · Authors · 2025-12-02
**Summary of Rebuttal Discussions**

# Summary of Rebuttal Discussions

We thank the reviewers for their constructive feedback and the AC(s) for their time. In the following, we provide a summary of the reviews, discussion, and improvements we have made on the paper, which we hope will ease the new AC in their assessment.

  1. **Recognized Strengths.** Reviewers originally acknowledged (i) the novelty of the intersection of long-term fairness and selective labels (5xkx, rN9Y); (ii) the strength (5xkx) and rigor (rN9Y) of our theoretical contributions; and (iii) that our claims are supported by empirical evidence (RQL1, xA3L).

  2. **Rebuttal Consensus.** Importantly, following the comprehensive updates detailed below, Reviewers RQL1 and xA3L confirmed that we had successfully addressed their questions and supported the acceptance of the paper. They joined Reviewer 5xkx, who was already supportive, resulting in a strong consensus before the system freeze.

  3. **Comments and Improvements.** We believe that the updates based on reviewers' comments have further increased the strength of our contributions. Next, we present the main comments and how we addressed them:

> **Comment:**  Assumption 1 might be too restrictive. (5xkx, rN9Y, xA3L)

**Update:** We evaluated a variant of SELLF where individuals with a score below 0.25 are always rejected, violating Assumption 1. The experiments in Sec. 5 and Appendix F.4 show that SELLF is able to ensure fairness and surpass baselines even with this violation. This suggests that, while the overlap assumption is a sufficient condition, the algorithm can be empirically robust even in threshold-based settings. Thus, an interesting direction for future work is to theoretically investigate this result by further studying the necessary conditions for the algorithm to provide guarantees.

> **Comment:** Renyi regularization should be better motivated, and the stability of IPW should be discussed. (5xkx, rN9Y)

**Update:** We updated Sec. 4 to better motivate the Renyi regularization (previously Renyi loss). This regularization explicitly penalizes the divergence between the distributions, thus preventing the policy from drifting into regions where propensity weights would explode, stabilizing IPW. In the new Appendix F.1, we also evaluated empirically the Renyi divergence during learning, which combined with the ablation study in Sec. 5.1 and the analysis in Appendix F.3, shows empirical evidence of the importance of the Renyi regularization, which in turn ensures the stability of the IPW estimates. Also, IPW is calculated taking into account the acceptance in all iterations, improving stability against IPW in a single iteration.

> **Comment:** Experiments should include additional baselines as ELBERT and FOCOPS. (5xkx, RQL1, xA3L)

**Update:** We implemented two extra baselines, ELBERT (for long-term fairness) [2] and FOCOPS [1] (constrained RL), with results presented at Sec. 5 and Appendix F.4. Our results confirm that because these methods ignore the partial observation of labels, they fail to minimize true disparity, where SELLF succeeds. Furthermore, we updated the related works (Sec. 1.1 and Appendix A) to discuss the missing citations.

> **Comment:** Experiments should include a non-linear predictor (RQL1, xA3L) and new scenario. (RQL1)

**Update:** In the new Appendix F.2, we discuss the impact of the dimension of the hypothesis space in our results, and display results with a nonlinear neural network as the predictor. In this configuration, SELLF was still able to surpass baselines in terms of fairness. Moreover, we introduced a third application scenario based on crime recidivism (COMPAS dataset) and using the accuracy parity fairness principle. In this configuration, SELLF obtained the best results in terms of fairness.

[1] Zhang, Yiming, Quan Vuong, and Keith Ross. "First order constrained optimization in policy space." Advances in Neural Information Processing Systems 33 (2020): 15338-15349.

[2] Xu, Yuancheng, et al. "Adapting Static Fairness to Sequential Decision-Making: Bias Mitigation Strategies towards Equal Long-term Benefit Rate." International Conference on Machine Learning. PMLR, 2024.

---

### Meta-Review · Area_Chair_gvdH · 2026-01-05

**Summary:**

I carefully reviewed the discussion between the authors and the reviewers during the rebuttal phase. Based on this, I infer that the final scores after rebuttal are likely around 6, 4, 6, and 6 (since the authors did not summarize the score changes before and after the rebuttal, this is only my estimation). Overall, three reviewers are generally positive, but none expressed a strong enthusiasm for acceptance, placing the paper in a borderline situation. Personally, I lean slightly toward rejection, because Reviewer rN9Y's concerns are not fully addressed.

**Reviewer Concerns:**

Reviewer RQL1 and Reviewer xA3L's concerns were addressed, but Reviewer rN9Y's concerns are still outstanding.

**Reviewer Scores:**

I think none reviewers would change their score.

---

### Decision · Program_Chairs · 2026-01-26

Reject